# Molecular insights into RNA recognition and gene regulation by the TRIM-NHL protein Mei-P26

Anna Salerno-Kochan[1,2] , Andreas Horn[3], Pritha Ghosh[4], Chandran Nithin[4] , Anna Kościelniak[1], Andreas Meindl[3], Daniela Strauss[3], Rościsław Krutyhołowa[1] , Oliver Rossbach[5], Janusz M Bujnicki[4,6], Monika Gaik[1] , Jan Medenbach[3] , Sebastian Glatt[1]

The TRIM-NHL protein Meiotic P26 (Mei-P26) acts as a regulator of cell fate in *Drosophila*. Its activity is critical for ovarian germline stem cell maintenance, differentiation of oocytes, and spermatogenesis. Mei-P26 functions as a post-transcriptional regulator of gene expression; however, the molecular details of how its NHL domain selectively recognizes and regulates its mRNA targets have remained elusive. Here, we present the crystal structure of the Mei-P26 NHL domain at 1.6 Å resolution and identify key amino acids that confer substrate specificity and distinguish Mei-P26 from closely related TRIM-NHL proteins. Furthermore, we identify mRNA targets of Mei-P26 in cultured *Drosophila* cells and show that Mei-P26 can act as either a repressor or activator of gene expression on different RNA targets. Our work reveals the molecular basis of RNA recognition by Mei-P26 and the fundamental functional differences between otherwise very similar TRIM-NHL proteins.

## Introduction

RNA-binding proteins (RBPs) play key roles in the post-transcriptional regulation of gene expression. They comprise a large and functionally diverse group of proteins involved in all aspects of RNA biology from RNA synthesis to its degradation. RBPs typically bind RNAs through dedicated RNA-binding domains (RBDs) (1). Several members of the evolutionary conserved TRIM-NHL family use their NHL domains to interact with RNA (2, 3, 4, 5, 6). The TRIM-NHL protein family shares a common architecture that comprises an N-terminal tripartite motif (TRIM, consisting of a RING domain, one or two B-Box type zinc fingers and a coiled-coil domain) followed by a C-terminal NCL-1, HT2A, and LIN-41 (NHL) domain (7, 8, 9, 10, 11). The NHL domain folds into a β-propeller that typically acts as a scaffold to mediate interactions with other biomolecules such as proteins, DNA or RNA (6, 12, 13).

TRIM-NHL proteins play important roles in development where they control cell fate decisions to regulate differentiation and cell growth (11, 14). The *Drosophila melanogaster* genome encodes several proteins with a TRIM-NHL-like architecture, among them Brain tumor (Brat) and Meiotic-P26 (Mei-P26). Mei-P26 was identified as a regulator of differentiation in the male and female germline and its loss results in over-proliferation of germline cells, tumor formation and sterility (15, 16, 17, 18, 19). Germline homeostasis depends on the maintenance of germline stem cells in the stem cell niche and on the proper differentiation of their progeny into gametes (20). In the female germline, Mei-P26 supports both cellular programs. It ensures maintenance of germline stem cells through control of BMP signaling (17), but also promotes differentiation of daughter cells upon exit from the stem cell niche (18). In the male germline, Mei-P26 limits mitotic divisions during the differentiation process of precursor germ cells preventing over-proliferation (16). Moreover, ovarian cells lacking Mei-P26 activity grow abnormally large and exhibit increased nucleolar size (21).

The function of the closely related Brat protein critically depends on its NHL domain as its deletion or other sequence alterations result in strong phenotypes that can be partially rescued by expression of the NHL domain alone (22, 23). The Brat NHL domain participates in multiple direct or RNA-mediated interactions involving factors such as Pum, eIF4EHP and Miranda as binding partners (23, 24, 25, 26, 27, 28, 29). Similarly, Mei-P26 function requires several additional proteins, among them Sxl (Sex-lethal), Bam (Bag of marbles), Bgcn (Benign gonial cell neoplasm), Wuho, and Ago1 (Argonaute-1) (17, 21, 30, 31). The Mei-P26 NHL domain is required for the interaction with at least some of these proteins, as mutations leading to substitutions in its sequence abolish the interaction with Ago1 and impair ovarian stem cell maintenance (21). Finally, the NHL domains of both Brat and Mei-P26 interact with RNA in a sequence-specific manner (3, 6, 32).

[1]Malopolska Centre of Biotechnology, Jagiellonian University, Krakow, Poland   [2]Postgraduate School of Molecular Medicine, Warsaw, Poland   [3]Biochemistry I, University of Regensburg, Regensburg, Germany   [4]Laboratory of Bioinformatics and Protein Engineering, International Institute of Molecular and Cell Biology, Warsaw, Poland   [5]Institute of Biochemistry, University of Giessen, Giessen, Germany   [6]Bioinformatics Laboratory, Institute of Molecular Biology and Biotechnology, Faculty of Biology, Adam Mickiewicz University, Poznan, Poland

Correspondence: monika.gaik@uj.edu.pl; Jan.Medenbach@vkl.uni-regensburg.de; sebastian.glatt@uj.edu.pl
Chandran Nithin's present address is Laboratory of Computational Biology, Faculty of Chemistry, Biological and Chemical Research Centre, University of Warsaw, Warsaw, Poland.

 

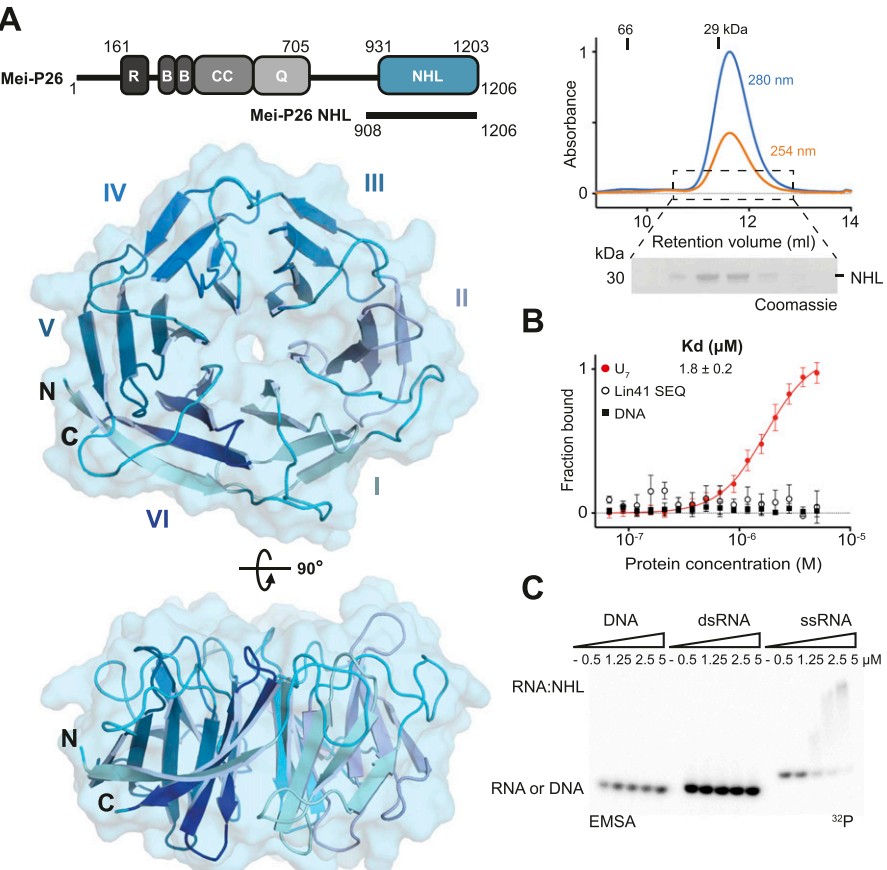

**Figure 1. Structure of the Mei-P26 NHL domain and its specific ssRNA recognition.**
**(A)** Mei-P26 domain organization with NHL domain shown in blue (top left). R, RING domain; B, B-Boxes; CC, coiled coil, Q, glutamine rich region. Size-exclusion chromatography profile and SDS–PAGE gel for Mei-P26 NHL protein with mass markers indicated at the top. Blue line on the chromatogram corresponds to 280 nm wavelength, orange line to 254 nm wavelength (top right). Cartoon and surface representation of the crystal structure of the Mei-P26 NHL domain in top surface and side orientations (encompassing amino acids 931–1,203) (left bottom). **(B)** Microscale thermophoresis results for the fraction of bound DNA, dsRNA, or ssRNA ($U_7$) ligands for different concentrations of Mei-P26 NHL. The dissociation constant ($K_d$) was calculated from at least three independent experiments (n ≥ 3). **(C)** EMSAs employing various concentrations of the Mei-P26 NHL domain (as indicated above each lane) on a single-stranded DNA oligonucleotide, an RNA hairpin structure (both as indicated in the microscale thermophoresis panel), or a single-stranded $U_9$ RNA oligonucleotide. Depicted is a representative gel of three independent replicates.
Source data are available online for this figure.

Considering the functional importance of the Mei-P26 NHL domain, we conducted comprehensive structural and functional analyses. We confirm the high structural similarity between the NHL domains of Mei-P26 and Brat, but also detect subtle differences that influence their function and affect target specificity. Computational modeling of the Mei-P26 NHL interaction with its RNA substrate allowed us to predict and experimentally validate key residues critical for sequence-specific RNA binding. Finally, we used individual-nucleotide resolution cross-linking and immunoprecipitation (iCLIP) to identify cellular mRNA targets of full-length Mei-P26 and its NHL domain. We confirm the recognition of several identified target sequences by Mei-P26 using individual electromobility shift assay (EMSA) analyses. Furthermore, we show that Mei-P26 directly affects the expression of reporters that carry these newly identified Mei-P26 binding sites and dissect the domain requirements for regulation. In summary, our results deepen the molecular understanding of NHL–RNA interactions that play a key function in governing cell fate decisions.

## Results

### The crystal structure of the *Drosophila* Mei-P26 NHL domain

To gain molecular insight into RNA binding of *D. melanogaster* Mei-P26, we aimed to obtain structural information of its C-terminal NHL domain (Mei-P26 NHL) at high resolution. Therefore, we predicted the beginning of the NHL domain and performed expression trials of different constructs in bacterial, yeast and insect cells. Only the baculovirus expression system allowed for the production and purification of the Mei-P26 NHL domain. We obtained large quantities of Mei-P26 NHL$_{aa\ 908–1206}$ producing a homogenous monomer of ~30 kD, which was completely free of any proteinaceous contaminants and nucleic acids (Fig 1A). Mei-P26 NHL crystallized in several tested conditions and we collected numerous complete datasets at various synchrotron sources. We solved the structure of the Mei-P26 NHL domain at 1.6 Å resolution by molecular replacement using the backbone of the previously determined Brat NHL domain (28) as a reference model. The N terminus of the NHL domain remains invisible because of averaging of its conformations throughout the crystal but is present as there was no indication of proteolytic degradation during purification and crystallization (Fig S1A and B). After refinement, an atomic model could be obtained with R/R$_{free}$ values of 20.6%/21.9% obeying all basic rules of protein stereochemistry (Table 1). Mei-P26 NHL folds into a six-bladed β-propeller with a donut-like shape and a diameter of ~45 Å and a height of ~25 Å (Fig 1A). The two molecules located in the asymmetric unit are connected by a di-sulfide bond between Cys1030, but the functional relevance of this dimerization under reducing conditions in vivo remains questionable. The six blades

**Table 1.  Data collection and refinement statistics.**

|  | Mei-P26 NHL, PDB ID 7NYQ, BESSY II, MX14-1 |
|---|---|
| Data collection |  |
| Space group | P2$_1$ |
| Cell dimensions |  |
| a, b, c (Å)[a] | 34.71, 116.50, 64.70 |
| α, β, γ (°) | 90, 96.362, 90 |
| Wavelength | 0.9184 |
| Resolution (Å)[b] | 43.17–1.6 (1.64–1.6) |
| $R_{meas}$ (%) | 7.7 (160.9) |
| $I/\sigma(I)$ | 10.15 (0.85) |
| $CC_{1/2}$ | 0.99 (0.46) |
| Completeness (%) | 98.8 (97.0) |
| Redundancy | 3.82 |
| Refinement |  |
| Resolution (Å) | 43.17–1.6 |
| No. reflections | 66,477 |
| $R_{work}/R_{free}$ | 0.196/0.215 |
| No. atoms |  |
| Protein | 8,923 |
| Ligand/ion | No ligands |
| Water | 320 |
| B factors |  |
| Protein | 36.7 |
| Ligand/ion | Not applicable |
| Water | 38.3 |
| r.m.s deviations |  |
| Bond lengths (Å) | 0.007 |
| Bond angles (°) | 0.689 |
| Ramachandran statistics (%) |  |
| Outliers | 0 |
| Allowed regions | 4.59 |
| Favored regions | 95.41 |
| Rotamer outliers (%) | 0.21 |
| Clash score | 3.02 |
| MolProbity score | 1.41 |

[a]Values in parentheses are for highest-resolution shell.
[b]Resolution limits according to I/σ of 2 is 1.75 Å.

are asymmetrically distributed in a radial fashion around a central axis. Each unit is composed of four antiparallel β-strands connected by loops of various length and degrees of flexibility. In the center of the molecule the six β-sheets form a solvent-filled channel with a diameter of ~12 Å. The overall structure is stabilized by a β-sheet complementation of the first N-terminal β-strand (aa 933–940) with the sixth sheet (aa 1178–1202) in a molecular velcro-like fashion.

## Mei-P26 NHL binds to single-stranded RNA

We compared the structure of Mei-P26 NHL to the NHL domains of Brat, Lin41 and Thin/Abba to gain further insights into their unique and commonly shared features (3, 32, 33). All four proteins act as post-transcriptional repressors of gene expression and recognize their target RNAs via their NHL domains. Recently, the RNA binding modes for Brat and Lin41 have been reported, highlighting the existence of two distinct recognition mechanisms. In detail, the NHL domain of Brat binds to linear single-stranded RNA motifs (6), whereas the NHL domain of Lin41 prefers RNA hairpins (3). The NHL domains of the *D. melanogaster* Mei-P26 and Brat are closely related to each other, whereas the NHL domain of Lin41 from *Drosophila rerio* noticeably resembles the NHL domains of the fly Wech and human TRIM71 proteins (3, 6). Thin/Abba is a member of a distinct branch of the TRIM-NHL phylogeny, which also includes *Homo sapiens* TRIM32 and *Caenorhabditis elegans* NHL-1 (14). These evolutionary relationships are also observed on the structural level, although all four NHL domains in principle share a similar overall architecture and differ only in the length of individual loops and β-strands (Fig S1C). As with Brat and Lin41, the NHL domain of Mei-P26 and Thin/Abba also exhibit a highly positively charged patch on their top surface (Fig S1D) (3, 32, 33, 34). Because it has been experimentally demonstrated that RNA binding in NHL domains occurs via this charged surface area, it was previously assumed that Mei-P26 NHL can also bind to RNA (3, 6, 32). However, 11 out of 15 of the residues important for RNA binding in the closely related Brat NHL domain are not conserved in the Mei-P26 NHL domain (Figs S1E and S2). There is, however, one notable exception: three amino acids important for the recognition of the first position of the RNA target by Brat (a uridine base interacting with Asn800, Tyr829, and Arg847) are conserved in Mei-P26 (Asn970, Tyr 999, and Arg1017). This suggests that Mei-P26 also binds a uridine base in the same region but uses varying, non-conserved amino acids to recognize different sequence motifs in its target RNAs.

To assess the binding specificity of the Mei-P26 NHL domain experimentally, we performed complementary microscale thermophoresis (MST) and EMSA experiments. MST experiments showed purified Mei-P26 NHL binds a single-stranded polyU RNA sequence (U$_7$, K$_d$ = 1.8 ± 0.2 $\mu$M), whereas no interaction could be detected with a folded RNA hairpin recognized by Lin41 (Lin41 SEQ) or a single stranded 7 nt DNA oligonucleotide (TTTTACA; Fig 1B). EMSA experiments confirmed the complex formation between Mei-P26 NHL and a poly-U RNA oligonucleotide at different salt concentrations without affecting RNA integrity (Fig S3A and B) and verified its inability to bind other tested nucleic acids (Fig 1C). The similarity between Mei-P26 NHL and Brat as well as the interaction data indicate that both proteins preferentially recognize linear motifs in single-stranded RNA (ssRNA) molecules.

## Mei-P26 NHL recognizes U-rich RNA sequences

To gain further insight into the sequence specificity of Mei-P26 NHL, we used recently developed algorithms (35) to re-analyze the data from RNAcompete experiments previously carried out by the Hughes and Morris groups (6). We sought to obtain a complete list

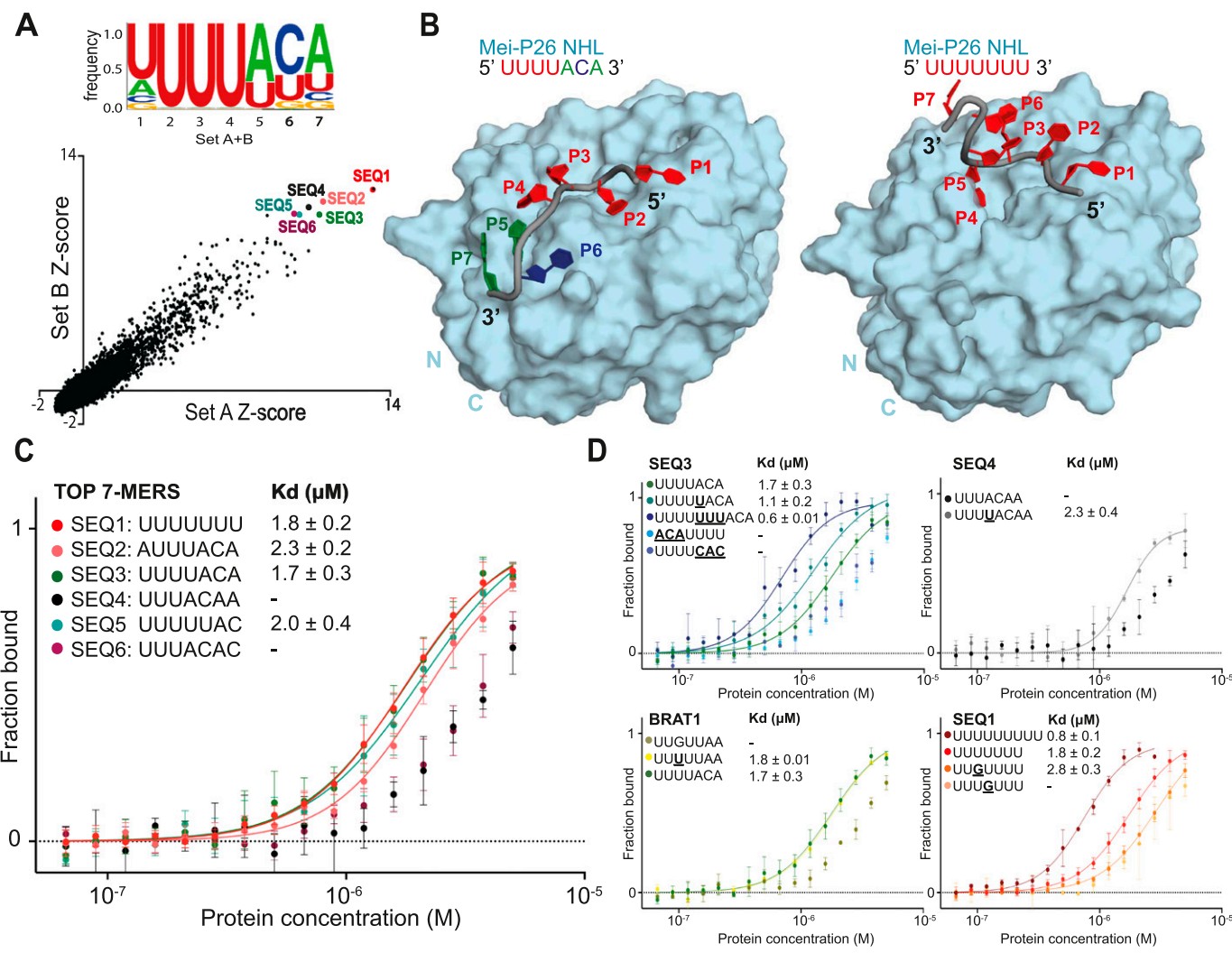

**Figure 2. The Mei-P26 NHL domain specifically recognizes single-stranded, U-rich RNA motifs.**
**(A)** Analysis of the Mei-P26 NHL RNAcompete data. Scatter plot representing Z-scores for two halves of the RNA pool (set A and set B) shown at the bottom; six highest scoring 7-mers were highlighted in color in right corner of the plot. The consensus motif derived from the indicated sequence motifs is indicated at the top. **(B)** Comparison between the Mei-P26 NHL in complex with UUUUACA (left) and UUUUUUU (right) (color-coded RNA sequences depicted at the top). The complexes were computationally modeled using molecular dynamics simulations by introducing oligonucleotides onto the surface of the Mei-P26 NHL domain. **(C, D)** Microscale thermophoresis–binding studies of the NHL domain to the six highest scoring 7-mer sequences (C), variants thereof, or the motif recognized by the closely related protein Brat (D). $K_d$ values at least three independent experiments (n ≥ 3). Microscale thermophoresis–binding curves for protein-oligonucleotide complexes with calculated $K_d$ were marked in solid lines, curves with $K_d$ > 3 µM were presented as experimental points.

of oligonucleotide sequences that may interact with the NHL domain of Mei-P26, as an initial step of defining Mei-P26 NHL targets. In our analyses, we used a previously generated library of almost all possible 7-mer oligonucleotide fragments that have been tested for binding to purified GST-tagged Mei-P26 NHL (6). The relative binding of each of the sequences is presented as a Z-score calculated from a dataset computationally separated into two subsets termed Set A and Set B and for a combined subsets A + B. Consensus motifs recognized by Mei-P26 NHL (Fig 2A) were calculated from the 10 top-scored sequences of 7-mer motifs. Of note, we performed the same control analysis for the Brat NHL domain and subsequently confirmed the binding of its RNAcompete-driven consensus motif (BRAT1: UUGUUAA) using MST (Fig S3C and D). Our data are consistent with previously published results (6) and show that Mei-P26

NHL preferentially binds oligonucleotides including three or more consecutive uridines commonly with additional adenine and/or cytosine residues at the 3' end of the sequence.

### Modeling of the NHL–RNA complexes

To better understand the mechanistic nature of RNA binding, we performed molecular dynamics (MD) simulations for the NHL domains of Mei-P26 and Brat with the U-rich RNA sequences derived from RNAcompete (Figs 2B, S4, and S5). Initial models of Mei-P26 in complex with RNA (SEQ1: UUUUUUU, SEQ3: UUUUACA, or BRAT1: UUGUUAA) were generated by modeling the RNA to reflect the conformation of the UUGUUGU nucleotide bound to the Brat NHL domain (PDB ID: 5EX7). According to our analysis, SEQ3 binds most

stably to the Mei-P26 NHL domain and converges into a well-defined conformation (Figs 2B and S4). Simulations with the top-scored polyU oligonucleotide (SEQ1) and the inverted SEQ3 sequence (ACAUUUU) resulted in much higher RMSD values, indicating a weaker fit in comparison to the SEQ3 RNA (Fig S4A). The top five clusters obtained by the MD simulations with SEQ3 are much more similar to each other than any of them compared with the clusters obtained with SEQ1. Moreover, we observed lower variance in the 3' region and more structural heterogeneity of the 5'-RNA docking site between the individual clusters of the Mei-P26:UUUUACA models (Fig S4B). Together, our simulations emphasize the importance of the ACA trinucleotide anchor adjacent to poly-uridine stretches for Mei-P26 RNA recognition (Figs 2B and S4). Although similar affinities were determined for both oligonucleotides (Figs 2C and S6), an elevated value for the polyU could have been indirectly caused by the so called avidity effect (36) (Fig 2D). This effect describes a scenario where an individual binding event increases the probability of additional interactions occurring in its close proximity leading to the enhancement of observed Kd value. Therefore, the avidity refers to the synergistic accumulation of multiple affinities, whereas the affinity itself describes a strength of a single interaction.

In summary, our in silico analysis, together with RNAcompete results, suggests that Mei-P26 NHL may use various mechanisms to interact with different RNA sequences despite similar binding constants. In detail, it seems to recognize specific sequence in ssRNA, but also may have ability to interact with U-rich sequences in rather unspecific manner. To obtain a relative comparison between models and crystal structures using the same modeling restraints, we performed three control MD simulations of the Brat NHL protein with both the Brat NHL and the Mei-P26 NHL top-scoring sequences from the RNAcompete experiment (BRAT1: UUGUUAA, SEQ3: UUUUACA) and the co-crystallized UUGUUGU nucleotide (PDB ID: 5EX7) (Fig S5A). Interestingly, the Brat NHL top-scoring sequence (BRAT1) from the RNAcompete experiment showed a relatively similar RMSD profile to the Mei-P26 NHL top-scoring sequence (SEQ3). Even if the model of crystallized Brat: UUGUUGU complex shows the lowest RMSD in comparison to the starting model, we still observed a slight structural divergence from the obtained crystal structure (Fig S5B). Strikingly, the MD simulations for both Mei-P26 NHL:SEQ3 and Brat NHL:BRAT1 complexes are of a similar range (Figs S4 and S5), but to show the atomic importance of individual residues the obtained models cannot substitute the crystal structures. Notably, the RMSD profile of Brat NHL:SEQ3 increases later during simulations, whereas the Mei-P26 NHL:SEQ3 complex remains stable, indicating that the ACA trinucleotide does not adapt well to the Brat NHL surface. During the simulation of the Brat-BRAT1 complex, the guanosine nucleotide remains in a stable interaction with the binding pocket further supporting its essential role in sequence-specific RNA recognition by Brat. Hence, our computational analyses support the premise that Mei-P26 and Brat have very distinct RNA recognition modes.

## Experimental characterization of the RNA target specificity of Mei-P26 NHL

Next, we set out to experimentally validate the target predictions and performed in vitro binding experiments using the six top ranked RNA sequences. Our results show that untagged Mei-P26 NHL interacts with four of the identified 7-nt RNA oligonucleotides at affinities that would be expected for specific but transiently associated RBPs (Figs 2C and S6). Among the identified sequences, Mei-P26 NHL showed the highest affinities towards SEQ3 (K$_d$ 1.7 $\mu$M) and polyU (K$_d$ 1.8 $\mu$M). Mei-P26 NHL also exhibited measurable affinities to the SEQ2 and SEQ5 RNA oligonucleotides but binding to SEQ4 and SEQ6 was reduced. Limited solubility of purified Mei-P26 NHL protein at higher concentrations did not allow determination of the dissociation constants for these oligonucleotides. These data demonstrate sequence-specific recognition and discrimination by the Mei-P26 NHL domain.

To gain further insights into the RNA sequence preferences of Mei-P26 NHL, we systematically altered individual positions and features of oligonucleotides identified as Mei-P26 NHL ligands. First, we added uridine nucleotides at the 5' end of SEQ3, which resulted in an increased binding affinity (Fig 2D). This observation agrees well with the finding that Mei-P26 NHL also displays an increased affinity towards extended polyU sequences (e.g., U$_9$; Fig 2D). Of note, NHL domains usually bind to relatively short RNA motifs and specifically recognize only a few nucleobases. For instance, the NHL domain of Brat establishes contacts with only six consecutive bases (6), whereas Lin41 only recognizes two nucleobases and makes most of its contacts via the ribose-phosphate backbone (3). Hence, we assume that the NHL domain of Mei-P26 also recognizes a short stretch of bases (Figs 2B and S4). Furthermore, we created an inverted SEQ3 oligonucleotide (ACAUUUU; Fig 2D), which exhibited decreased affinity and indicated the necessary 5' to 3' directionality of the motif. Analysis performed with a UUUUCAC sequence also displayed significantly lower affinity further indicating the importance of the ACA anchor (Fig 2D). We also managed to convert a poor substrate (UUUACAA) into a substrate able to be bound by the addition of a single U nucleotide (**U**UUUACAA), highlighting the necessity of at least four consecutive uridines. Conversely, insertion of a guanosine nucleotide at asymmetric and symmetric positions into the polyU sequence (SEQ1) reduced the affinity to Mei-P26 (Fig 2D). Extending SEQ1 by two additional uridines, lead to the observation of increased affinity, illustrating the aforementioned avidity effect for polyU sequences (36) (Fig 2D). In summary, our results show sequence-specific RNA recognition by the Mei-P26 NHL domain and confirm computational predictions based on the RNAcompete data.

## Identification of key residues involved in RNA recognition

The comparison of the Mei-P26 and Brat NHL domain structures prompted us to analyze the key features of these domains involved in sequence-specific RNA recognition. The models of the Mei-P26 NHL domain in complex with different RNA ligands (Fig S4) allowed high confidence prediction of surface areas and individual amino acids important for RNA binding. We predicted Arg1017 to be in close proximity to the first base (U1) that could stack with Y999 (Fig 3A). Lys1172 and Arg1175 are predicted to be in hydrogen bonding distance to the fourth and sixth base (U4, C6) of the RNA substrate, respectively. Moreover, in our MD experiment for Mei-P26 with SEQ3, Arg1175 was predicted to interact with the second, third, fourth and fifth base (U2, U3, U4, and A5), whereas Arg1150 is likely to interact

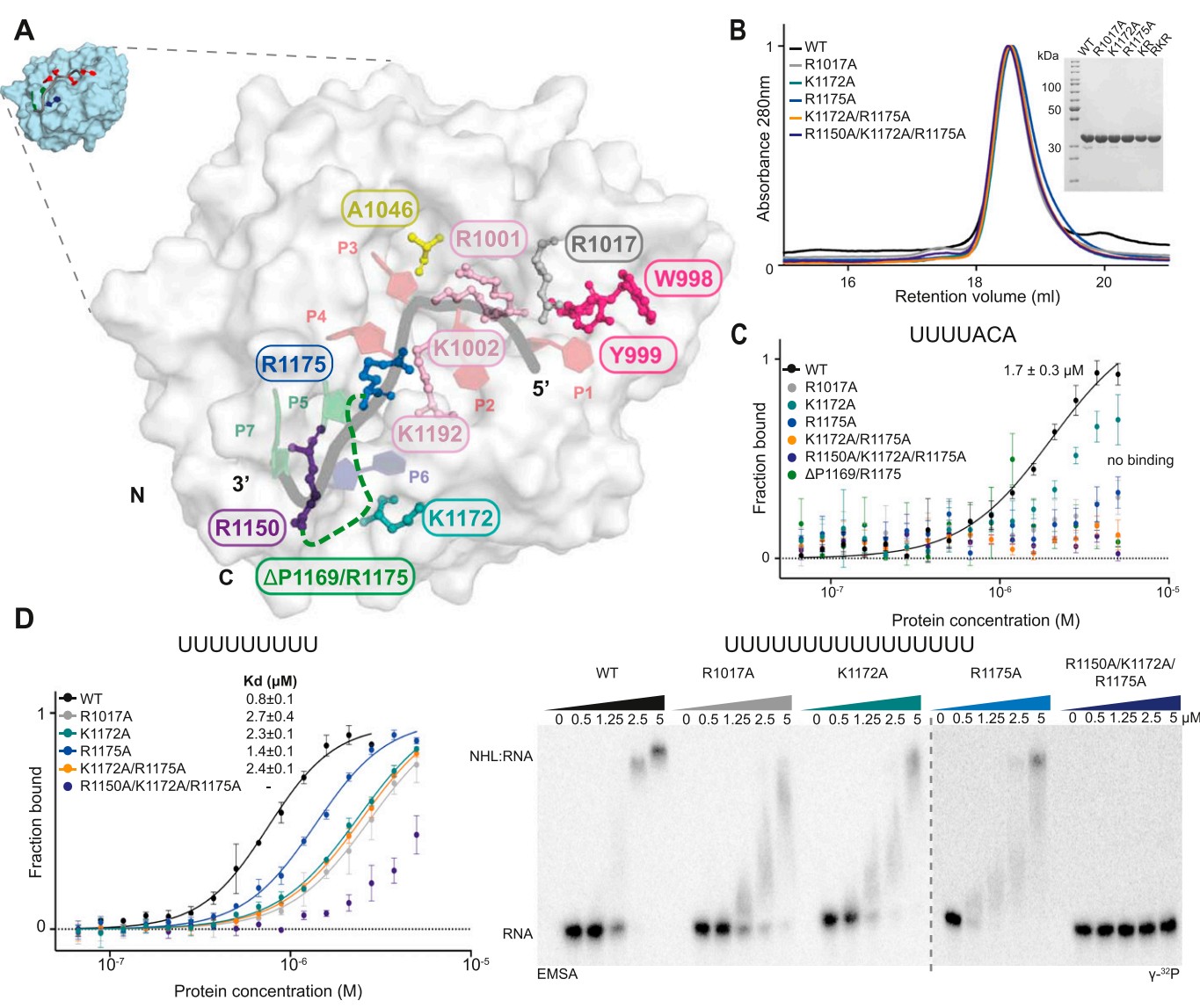

**Figure 3. Identification of amino acids in the NHL domain important for RNA binding.**
**(A)** Model of the Mei-P26 NHL domain in complex with a UUUUACA RNA sequence. Select amino acids predicted to be in close proximity to the RNA are highlighted. **(B)** Size exclusion chromatography profile and SDS–PAGE gel for selected Mei-P26 mutated variants. Abbreviations for K1172A/R1175A (KR) and R1150A/K1172A/R1175A (RKR) are used in the inset. **(C, D)** Microscale thermophoresis (panels C and D [left]) or EMSA-based (panel D [right]) in vitro binding analyses of the recombinant NHL domain or variants thereof (as indicated in each graph) to RNA oligonucleotides with the sequence UUUUACA (C), U$_9$ (D [left]), or U$_{16}$ (D [right]). Dissociation constants (K$_d$) were calculated from three independent experiments (n = 3).
Source data are available online for this figure.

with A5, C6, and A7. Importantly, for Mei-P26 simulations with polyU, we did not observe an interaction of K1172 and R1150 with the RNA, suggesting that they might be involved in the recognition of the ACA anchor of the target RNAs. The docking analyses both emphasize the importance of the positively charged surface patch for RNA binding and highlight the potential contributions of three flexible loop regions that contain several positively charged residues. To confirm this hypothesis, we generated variants of Mei-P26 NHL carrying alanine substitutions of the individual residues (Mei-P26 NHL$_{Y999A}$, Mei-P26 NHL$_{R1017A}$, Mei-P26 NHL$_{K1172A}$, and Mei-P26 NHL$_{R1175A}$), combinations thereof (Mei-P26 NHL$_{K1172A/R1175A}$, Mei-P26 NHL$_{R1150A/K1172A/R1175A}$ [Mei-P26$^{RKR}$]) and deletions of the longest identified loop region (Mei-P26 NHL$_{\Delta P1169-R1175}$). Finally, we also substituted core residues in the center of the positively charged cavity (Mei-P26 NHL$_{R1001A/K1002A/K1192A}$). All variants were individually expressed in insect cells, purified and subjected to thermal shift assays to confirm proper folding and stability. With the exceptions of substitutions of Tyr999 (Mei-P26 NHL$_{Y999A}$) and residues in its proximity Arg1001 and Lys1002 (Mei-P26 NHL$_{R1001A/K1002A/K1192A}$), which strongly destabilized the domain or led to protein degradation (Fig S7A and B), all variants showed purity and stability parameters comparable to the wild-type Mei-P26 NHL protein (Figs 3B and S7A and B). We tested all stable Mei-P26 NHL variants in RNA interaction assays and observed severely

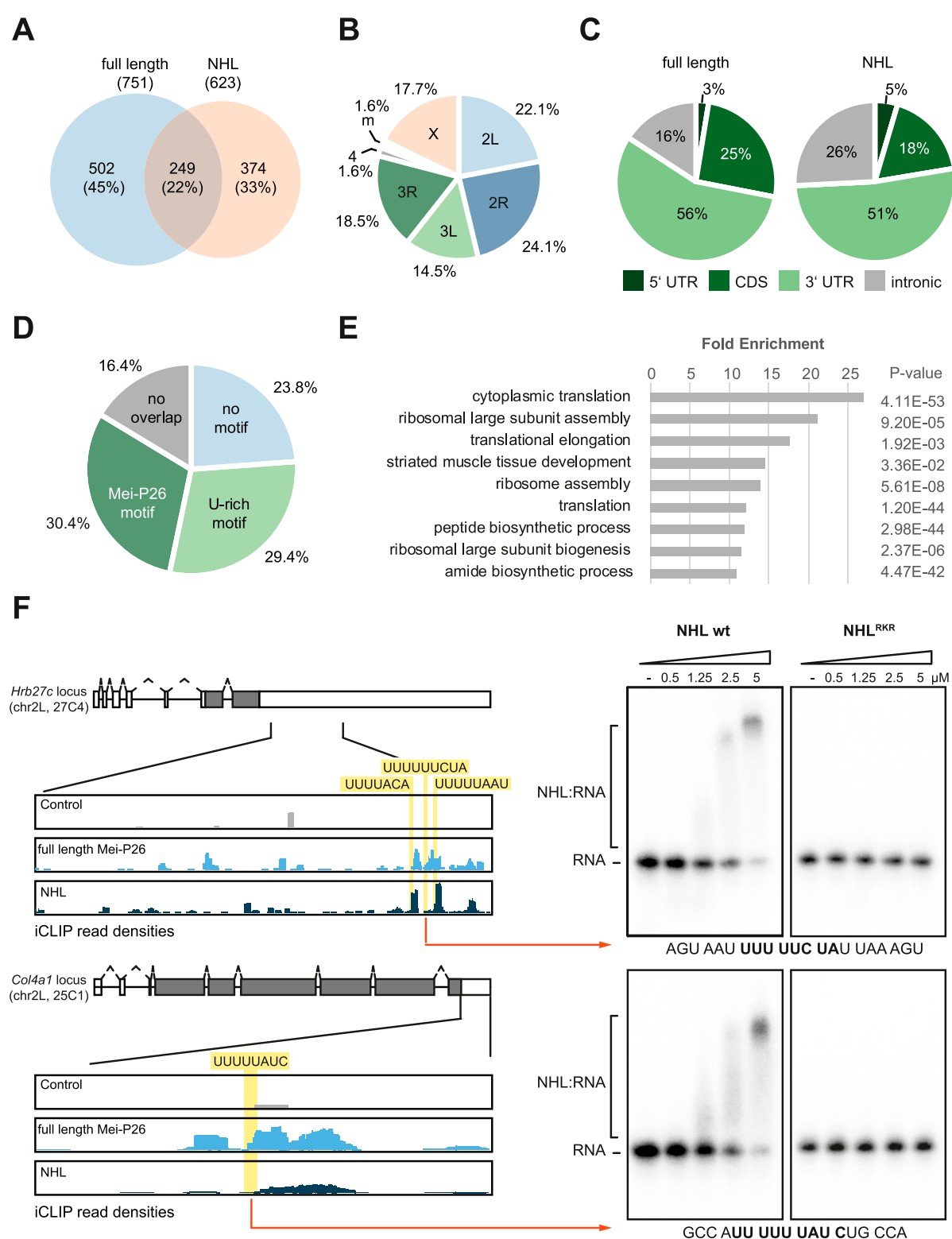

**Figure 4. Identification of Mei-P26 target RNAs in cultured cells.**
**(A)** Comparison of iCLIP data derived from either full-length Mei-P26 protein (depicted in blue) or its NHL domain (depicted in salmon). Number of genes containing a statistically significant local enrichment of cross-link positions (cross-link peaks) and percent values are given for each fraction. **(B)** Chromosomal origin of the 249 shared target genes identified in both iCLIP datasets. **(C)** Location of the full-length Mei-P26 (left) or its NHL domain-derived (right) iCLIP clusters within transcripts. **(D)** Presence of potential Mei-P26 RNA motifs in vicinity of the cross-link peaks in the shared target genes. No overlap: cross-linking of the full-length protein and the NHL domain occurred in different regions of the gene locus. **(E)** Gene Ontology term analysis of biological processes enriched in the shared target genes. **(F)** Validation of select

compromised or even complete loss of complex formation with SEQ3 (Fig 3C). To understand whether some of the variants exhibit different motif and sequence specificity, we also tested their binding to a polyU oligonucleotide ($U_9$ and $U_{16}$). Most of the variants still associate with the polyU oligonucleotides, albeit with reduced affinity (Fig 3D). Of note, our MD simulation indicated that Arg1175 participates exclusively in SEQ3 binding, but appears dispensable for polyU binding. The substitution of Arg1175 indeed strongly reduces Mei-P26 NHL ability to recognize the SEQ3 sequence, but shows only minor influence on its interaction with $U_9$. In contrast, residue Arg1017 that seems to stabilize U1 in both simulations likewise strongly contributes to the binding of both, SEQ3 and $U_9$. These data experimentally indicate that Mei-P26 might use different modes of RNA recognition for the association with different RNA motifs. Furthermore, we show that simultaneous substitution of residues R1150A, K1172A and R1175A fully abolishes the binding to any of the RNA sequences tested. In addition, in MD simulations performed for Mei-P26 NHL$_{R1150A/K1172A/R1175A}$ with $U_7$ sequence the unstable binding where observed, whereas in the case of UUUUACA sequence, the ACA part was poorly fitted into the protein binding pocket (data not shown). Therefore, our findings are consistent with the MD simulations and conclude that R1150 and K1172 residues are responsible for anchoring the ACA trinucleotide, whereas R1175 stabilizes the uridine tract. In summary, our results identify particular amino acid residues critical for RNA recognition.

We also tested binding of Mei-P26 NHL to the consensus Brat recognition sequence (UUGUUAA, BRAT1), which we obtained from the RNAcompete data (Fig S2) and which we used in our MD simulations (Fig S4). Whereas this RNA was only weakly bound by Mei-P26, a single substitution from G to U (BRAT**mut**UU**U**UUAA; Fig 2D) generated a Mei-P26–like sequence motif and allowed binding. Similarly, in our experiments, Brat NHL weakly interacts with Mei-P26 SEQ3, demonstrating the specificity of its NHL domain (Fig S3A). In the NHL domain of Brat, a single-point substitution (R875A) located in the center of the positively charged surface area impairs binding to the target sequence in the 3′ UTR of the *hunchback* (*hb*) mRNA (32). In Mei-P26, the corresponding residue at the same position is already an alanine (Ala1046) (Fig S8A and B). In an attempt to mimic Brat, we replaced the alanine with arginine (A1046R) in the Mei-P26 NHL domain. The substitution resulted in an overall destabilization of the domain and its failure to bind RNA (Figs S7B and S8C).

### Identification of Mei-P26 target mRNAs by iCLIP

To identify cellular RNA targets of Mei-P26, we performed individual-nucleotide resolution UV crosslinking and immunoprecipitation (iCLIP2 (37)) experiments. As *Drosophila* tissues that express Mei-P26 are not readily accessible for biochemical experimentation

such as iCLIP, we turned to cultured *Drosophila* Schneider 2 (S2) cells as a model system. S2 cells have been frequently and successfully used to study gene regulation processes that are operating in select, specialized tissues or cells (such as the germline or neurons), or that occur during specific developmental stages (6, 38, 39). Moreover, these cells have served to identify the mRNA targets of numerous RNA-binding proteins using CLIP or related methodology (40, 41, 42).

S2 cells express only low levels of mRNAs encoding Mei-P26 (43), suggesting that endogenous Mei-P26 protein is not abundant. To overcome this limitation, we transfected constructs encoding FLAG-tagged proteins as bait for the iCLIP experiments. We used full-length Mei-P26 protein, its NHL domain (aa 908–1,206), or the respective derivatives thereof that carry substitutions that impair RNA binding (R1150A, K1172A, and R1175A, Fig S9A). As expected, compared with the wild-type counterparts, variants with substitutions immunoprecipitate strongly reduced amounts of RNA (Fig S9B).

Analyses of the iCLIP data from the wild-type proteins identify a local enrichment of cross-link positions in 751 protein-coding genes for the full-length protein and 623 for the NHL domain that do not exhibit a bias regarding their chromosomal origin (Figs 4A and B and S9C and Supplemental Data 1). For both proteins, cross-linking mostly occurs at sites located in the 3′ UTRs of the target genes (Figs 4C and S9C). When comparing the iCLIP datasets obtained for full-length Mei-P26 and the NHL domain, only a moderate overlap is observed (22.1%, 249 loci, Fig 4A and Supplemental Data 2). In 71.9% of the shared target genes, crosslinking is observed in comparable positions in the gene body (at a distance of 50 nt or less, Fig 4D). After removal of contaminating sequences (e.g., mitochondrial transcripts, sno-RNA-derived reads, vector-derived sequences), the remaining 214 mRNA targets that are bound by the full length protein and the NHL domain were analyzed for the occurrence of Mei-P26 binding motifs in a region encompassing 30 nts upstream and 20 nts downstream of the crosslink positions. This revealed the presence of U-rich motifs in proximity to the respective crosslink peaks in 128 mRNAs (59.8%), 65 (30%) of which represent bona fide Mei-P26 target sequences (Fig 4D and F and Supplemental Data 2). Hence, in proximity to the iCLIP peaks, bona fide Mei-P26 motifs occur at a ~ninefold higher frequency than expected by chance ($f$ = 3.4% expected in random 50mer RNA fragments), validating the computational and experimental in vitro binding studies. Further analyses of the mRNAs that are bound by both, full-length Mei-P26 and its NHL domain, reveal an enrichment of genes encoding ribosomal proteins and translation factors (Fig 4E), hinting at a potential function of Mei-P26 in the control of the translation machinery which was previously proposed (21). Several of the newly identified, potential RNA targets from the iCLIP analyses were chosen for validation experiments using the recombinant Mei-P26 NHL domain. In all cases, EMSA experiments confirmed the interaction

---

Mei-P26 binding sites identified by iCLIP. Top left: schematic depiction of the *Hrb27c* and *Col4a1* gene loci. Introns are depicted as lines, exons as boxes; the grey shading indicates the protein coding region. Below: iCLIP read depth analyses of 3′ UTR regions (as indicated by the solid black lines) from experiments performed with either the full-length Mei-P26 protein (light blue), its NHL domain (dark blue), or from control experiments (grey). Potential RNA motifs recognized by Mei-P26 are highlighted in yellow. Right: EMSA analyses using RNA fragments derived from the iCLIP clusters (as indicated by the red arrows, sequences provided at the bottom of the gels), using different concentrations of the recombinant Mei-P26 NHL domain or its mutant derivative (Mei-P26$^{RKR}$: R1150A, K1172A, and R1175A, as indicated above the gels). Free RNA probe and NHL:RNA complexes are indicated on the left

Source data are available online for this figure.

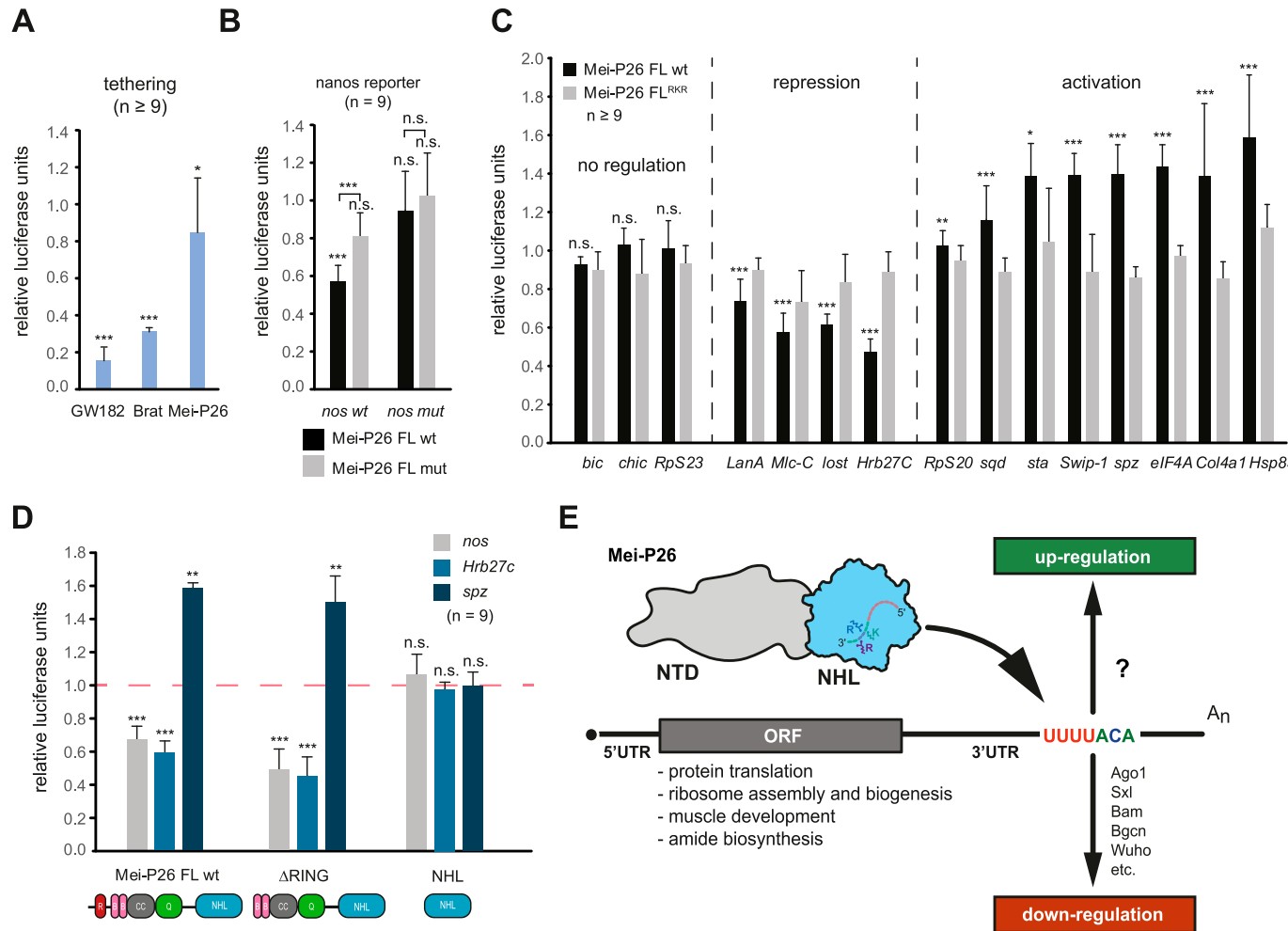

**Figure 5. Mei-P26 regulates gene expression via 3′ UTR binding sites.**
**(A)** Tethered function assay using Brat, Gw182 (positive control) and full-length Mei-P26. Activities are calculated relative to proteins that lack the tag for tethering.
**(B)** Mei-P26–mediated repression of a reporter that either bears the *nanos* mRNA 3′ UTR (*nos* wt) or a version thereof in which a U-rich sequence element previously implicated in regulation was mutated (*nos* mut). Reporter activity was determined in the presence of co-expressed wild-type Mei-P26 protein (grey bars), or a Mei-P26 protein carrying substitutions that affect RNA interaction. (Mei-P26[RKR]:R1150A, K1172A, R1175A; black bars). All activities are expressed relative to control reactions without overexpression of Mei-P26 (using an empty plasmid). **(C)** Reporter assays using 3′ UTR sequences derived from various Mei-P26–bound genes (as indicated at the bottom) and uisng wt Mei-P26-FL protein (grey bars) or its mutant derivative (black bars) as described for panel (B). All activities are expressed relative to control reactions without overexpression of Mei-P26 (using an empty plasmid). **(D)** Reporter assays using 3′ UTR sequences of *nos* (grey), *Hrb27c* (blue) and *spz* (dark blue) derived from Mei-P26-FL protein or its mutant derivatives (as indicated at the bottom) as described for panels (B and C). **(A, B, C, D)** For panels (A, B, C, D) mean values ± SD are depicted of at least three independent biological experiments performed in three technical replicates each. *P*-values were calculated with a two-sided *t* test relative to the control reactions described for each panel; n.s., not significant, *$P < 0.05$, **$P < 0.01$, ***$P < 0.001$. **(E)** Schematic depiction of Mei-P26–mediated post-transcriptional regulation of gene expression. Mei-P26 (NTD in grey, NHL domain in blue, mutated residues that are crucial for RNA binding are highlighted) associates with U-rich RNA motifs present in the 3′ UTRs of its RNA targets (depicted schematically) to regulate their expression. Regulation involves additional factors such as Ago1, Sxl, Bam, Bgcn, and Wuho that have been implicated in Mei-P26–dependent repression, whereas potential co-factors involved in Mei-P26–dependent gene activation remain to be identified.

and again the Mei-P26 NHL[RKR] mutant did not associate with the RNAs (Figs 4F and S10).

### The NHL domain is important for Mei-P26 gene-regulatory activity

To better understand the impact of Mei-P26 on gene expression, we conducted a series of reporter assays in cultured *Drosophila* Schneider 2 cells. We first used the phage-derived lambda-boxB system to artificially recruit full-length Mei-P26 to an RNA. For this, the lambda bacteriophage antiterminator protein N (lambdaN) was fused to Mei-P26 and two control proteins (Brat and GW182) to

tether them to a firefly reporter RNA that contains within its 3′ UTR several copies of the lambdaN binding site (boxB). The same proteins without the lambda peptide served as controls. In addition, a co-transfected plasmid encoding a renilla luciferase mRNA that lacks the boxB elements was used for normalization. In this experimental setup, Brat and the positive control GW182, which is involved in miRNA-mediated gene silencing, convey robust repression of the firefly luciferase reporter mRNA (32). In contrast, Mei-P26 exhibits only a weak gene regulatory activity (Fig 5A).

Next, we analyzed expression of a reporter RNA that contains a fragment of the *nanos* 3′UTR, a genetically identified target of

Mei-P26 (18). Co-expression of full-length Mei-P26 resulted in silencing of the reporter relative to a control mRNA that bears a 3′ UTR from an unrelated RNA (*msl-2*). Repression is dependent on the ability of Mei-P26 to bind to RNA through its NHL domain since the RKR substitution abolishes regulation (Fig 5B). A previously identified U-rich RNA sequence element in the 3′UTR of *nanos* mRNA (30) is critical for Mei-P26–mediated repression as its deletion completely abrogates regulation of the reporter (Fig 5B).

Using a similar experimental setup, we tested a series of selected mRNA targets that we identified in our iCLIP analyses (Fig 5C). To cover a broad selection of features, we selected from the set of genes that exhibited overlapping iCLIP peaks candidates that contained in vicinity to the crosslink position either (a) an identifiable Mei-P26 target motif (eIF4A, Rps20, Rps23, sqd, and Swip-1), (b) a U-rich sequence (bic, chic, and sta), or (c) no such element (Hsp83 and spz). In addition, all 3′ UTR sequences were included for which we had experimentally confirmed interaction with Mei-P26 (Col4A1, Hrb27c, LanA, lost, and Mlc-c, Figs 4F and S10 and Supplemental Data 2). In functional assays, the effect of Mei-P26 on the expression of the reporters was diverse. For instance, we could not detect a significant change to the expression of reporters that bear the 3′UTRs of *bic*, *chic* and *RpS23*. In contrast, *LanA*-, *Mlc-c*-, *lost*-, and *Hrb27C*-derived reporters exhibited significant repression. Unexpectedly, *RpS20*-, *sqd*-, *sta*-, *Swip*-1, *spz*-, *eIF4A*-, *Col4A1*-, and *Hsp83*-derived reporters showed Mei-P26–dependent activation (Fig 5C).

To further dissect the functional domain requirements of Mei-P26, we analyzed the regulatory activity of mutant derivatives. Substitutions in the NHL domain of Mei-P26 that abrogate RNA binding activity severely blunted regulation of all reporter RNAs, underlining the functional importance of the domain for activity (Fig 5C). Despite being expressed at a comparable level (Fig S11), a construct encompassing only the NHL domain was not sufficient for regulation (Fig 5D), demonstrating that additional sequences outside the NHL domain are required. Previously, it has been debated whether the N-terminal RING domain and its ubiquitin ligase activity contribute to the gene regulatory activity of Mei-P26 (21). However, deletion of the N-terminal RING domain neither abolished activation of *spz*, or repression of *nos* or *Hrb27c* reporters, demonstrating that ubiquitin ligase activity is dispensable in this experimental setup (Fig 5D).

**Mei-P26 has a well-defined dimerization interface**

To get additional insight into the possible Mei-P26 mRNA regulatory mechanism, we modeled Mei-P26 full-length protein structure using an artificial intelligence-based AlphaFold 2 (AF2) approach (44). The generated model showed a largely unstructured N-terminal region (aa 1–161), followed by a well folded RING-BBox-CC motif, an unstructured central part of the protein (Q-rich region) and a fully folded NHL domain at the C-terminus (Fig S12A). Of note, the obtained model of Mei-P26 domain alone is almost identical to our experimentally determined crystal structure (Fig S12B).

Previously it was noted that many TRIM proteins dimerize to perform their functions. Hence, we used AF2 to test if different regions in Mei-P26 show a potential to form homodimers. Therefore, we split the Mei-P26 sequence into two parts (Mei-P26$_{1-610}$ and Mei-P26$_{540-1206}$) and performed an AlphaFold 2 Multimer (45 Preprint) analysis with two copies of each variant as a template (Fig S12C). The computational model suggested a possible dimerization of Mei-P26 via a previously unannotated N-terminal region. This model is supported by a predicted helix-to-helix interaction between two antiparallel Mei-P26 helical motifs (aa 363–524) from separate molecules (Fig S12D). In summary, the obtained models suggest that Mei-P26 may dimerize, which presumably impacts on its mRNA recognition and regulatory ability. Furthermore, we have analyzed the presence of similar dimerization domains in other members of the TRIM-NHL family and found indication for similar motifs in Brat, Lin41 and Abba/Thin (Fig S12E).

## Discussion

TRIM-NHL proteins are required for proper development in metazoans and their NHL domains are crucial for function (25, 32, 46, 47). Despite their evolutionarily conserved architecture, NHL domains exhibit clear differences in RNA binding, recognizing diverse RNA sequences or RNA hairpin structures (3, 6). A structural comparison of the NHL domains of Brat, Lin41, Thin/Abba and Mei-P26 reveals similarities between Brat and Mei-P26 regarding their interactions with ssRNA, whereas Lin41 uses a different mode of interaction. Although none of our numerous attempts yielded co-crystals of the Mei-P26 NHL domain in complex with its RNA substrate, computational modeling allowed us to identify and experimentally validate key amino acid residues involved in RNA recognition. For instance, Mei-P26 and Brat use an evolutionary conserved interaction site to specifically recognize a uridine base in the first position of their RNA target. In contrast, recognition of other bases differs between the two proteins and they use different amino acid residues for substrate binding, resulting in different RNA specificity. Whereas Brat preferentially associates with a UUGUUGU RNA sequence, Mei-P26 recognizes a linear UUUUACA core motif indicating at least two distinct interaction modes between these two similar NHL domains.

Typically, individual RNA-binding domains recognize short RNA motifs of three to five nucleotides in length with rather moderate affinities (48). To increase affinity and specificity, often multiple binding domains are combined, either in tandem in the same polypeptide, or *in trans* through formation of protein complexes (49). Our measured affinities for Mei-P26 and Brat NHLs with short oligonucleotide sequences are in the low micromolar range. Thus, they are lower than the values previously described for Brat NHL with longer oligonucleotide sequences (6), but comparable with the ones obtained for Brat NHL bound to 6-mer oligonucleotide (29) and to those reported for CeLin41 NHL for shorter hairpins (3). We provide experimental evidence (Fig S9B) that in Mei-P26 protein regions outside the NHL domain also contribute to RNA target recognition/binding and that Mei-P26 acts as a dimer. In vivo, it is most likely that additional protein partners contribute to stable complex formation between Mei-P26 and its RNA targets. Genetic experiments demonstrated that besides Mei-P26, the proteins Sxl, Bam, Bgcn, and Wuho are crucial for the repression of Nanos protein production and differentiation of ovarian stem cells in the female germline (19, 30, 31, 50, 51). For regulation, these proteins likely form a large repressor

complex on the 3′UTR of *nos* mRNA (18, 50). Genetic ablation of any one of these proteins impaired regulation and resulted in strong phenotypes (19, 31, 52, 53, 54, 55, 56), demonstrating that these factors need to act jointly to achieve their function.

Moreover, we find surprising evidence that Mei-P26 not only functions in the repression of selected target mRNAs but acts as an activator on other mRNAs (Fig 5C). How can these two seemingly different activities be explained? In tethering experiments, where the closely related protein Brat acts as a strong silencer of gene expression, Mei-P26 exhibits only weak activity. This suggests that either tethering disturbs its function or Mei-P26 itself is not a strong regulator of gene expression. In the latter scenario, Mei-P26 might function in promoting complex assembly, recruiting other factors that act in gene regulation. Previously, Ago1, Sxl, Bam, Bgcn, and Wuho have been identified as co-repressors that act in concert with Mei-P26 in post-transcriptional regulation of gene expression (Fig 5E) (18, 19, 21, 31, 57). Potential activators that can be recruited by Mei-P26 for gene regulation remain to be identified.

Previously, it has been speculated that the N-terminal RING finger ubiquitin ligase domain is important for Mei-P26–mediated regulation of gene expression by promoting the turnover of RNA regulatory factors (21). Our data demonstrate that the Mei-P26 RING domain is not required for the regulation of the analyzed reporter RNAs. Similarly, the closely related protein Brat, a potent post-transcriptional regulator of gene expression, has a truncated TRIM domain that lacks the RING motif. However, in contrast to Brat, where the NHL domain alone was able to provide most of the activity of the full-length protein (22, 23), the isolated NHL domain of Mei-P26 does not exhibit gene regulation in functional assays (Fig 5D).

iCLIP experiments allowed us to identify and validate Mei-P26 binding sites in numerous transcripts. However, the S2 cells that we used for our experimentation express only a limited set of mRNAs, lacking most germline- and neuron-specific transcripts. Our analysis is thus limited to the identification of Mei-P26 binding sites present in the repertoire of expressed RNAs. Moreover, interacting partners likely shape the interaction profile of Mei-P26 in vivo. In *Drosophila* this has been well documented and analyzed in molecular detail for the protein Upstream of N-ras (Unr). In female flies, Unr is recruited to the *msl-2* mRNA by the protein Sxl through highly synergistic interactions (58, 59, 60, 61). It is expected that in vivo, the interaction of Mei-P26 with its RNA targets is further influenced by additional RNA binding proteins that interact with Mei-P26 (such as e.g. Sxl, Bam or Bgcn). In addition, other proteins interacting with Mei-P26 might modulate its activity and other factors might compete for the same or overlapping binding sites on RNA molecules.

Despite these limitations, our experimentation allowed us to identify numerous transcripts that encode translation factors and ribosomal proteins among the Mei-P26 target mRNAs. Previously, Mei-P26, like Brat, has been broadly implicated in the regulation of ribosome biogenesis by controlling the expression of *Myc* (21, 62, 63), which stimulates the expression of the Pol I transcriptional machinery thus promoting ribosome biogenesis (21, 64, 65). The coordinate regulation of ribosomal proteins and ribosomal RNA (rRNA) synthesis by Mei-P26 might allow it to efficiently tune ribosome biosynthesis which is linked to cell growth and the switch between proliferation and differentiation.

In summary, our computational, in vitro and in vivo studies provide a comprehensive insight into RNA recognition by Mei-P26 and reveal differences to the closely related protein Brat. We further identify and validate numerous novel mRNA targets of Mei-P26 and provide unexpected evidence that it can function in both repression and activation of gene expression.

# Materials and Methods

## DNA constructs

First, the coding sequence of Mei-P26 NHL (isoform C/E) was amplified by RT-PCR from total RNA (prepared from *Drosophila* embryos, strain Oregon-R) and cloned into a modified pFastBac HTA vector carrying an additional N-terminal GST tag sequence followed by a TEV cleavage site. Mutated variants of Mei-P26 NHL were obtained using the QuickChange cloning protocol. For Brat-NHL, the His6-ubiquitin fusion protein produced from the pHUE vector was used (a gift from I Loedige) (6). For the luciferase reporter plasmids, 3′ UTR fragments of select genes were RT-PCR amplified from total RNA prepared from *Drosophila* S2R+ cells (primers and RNA regions are provided in Table S1) and ligated into a modified pCasPeR-Heatshock vector containing a Firefly luciferase open reading frame (pHS-FLuc (39)) using the *Hpa*I and *Bgl*II restriction sites. A vector encoding Renilla luciferase (pHS-$B_m$-RLuc-(EF)m-SV40 aka pHS-RL) (39) was used for co-transfection and served as a reference for normalization. Plasmids encoding HA-tagged or lambda N-HA-tagged, full-length Mei-P26 or Brat were kindly provided by I. Loedige (MDC Berlin) and G. Meister (University of Regensburg). For iCLIP experiments, the coding sequences of Mei-P26 and its NHL$_{908-1206}$ domain were subcloned into a modified pAc5.1 vector using the *Sbf*I and *Not*I restriction sites to generate expression plasmids encoding N-terminally 2x FLAG-tagged proteins. Point mutations in the NHL$_{908-1206}$ domain were subsequently introduced using site-directed mutagenesis. The sequences of all oligonucleotides used for cloning are provided in the table (see Table S1). All vectors were validated by sequencing.

## Protein expression and purification

Mei-P26 NHL$_{908-1206}$ and its mutated variants were obtained using Bac-to-Bac baculovirus expression system according to standard protocols. For protein expression, Hi5 insect cells were infected at a MOI of 0.5 and grown for 3 d at 27°C. After harvesting (4°C, 15′, 7,000 rcf), the cell pellets were resuspended in lysis buffer (50 mM Hepes pH 7.5, 600 mM NaCl, 10% Glycerol, 5 mM MgCl$_2$, 1 mM DTT, DNase), snap frozen in liquid nitrogen and stored in −80°C. For purification, cells were lysed by three cycles of freezing and thawing, followed by mild sonication and centrifugation (4°C, 1 h, 80,000 rcf). The supernatant was applied and circulated on a Glutathione Sepharose 4 Fast Flow 16/10 column (GE Healthcare) for 16 h, then washed with buffer A (50 mM Hepes pH 7.5, 600 mM NaCl, 10% Glycerol, 1 mM DTT) and high salt buffer A (50 mM Hepes pH 7.5, 1 M NaCl, 10% Glycerol, 1 mM DTT). Elution occurred in a buffer containing 50 mM Hepes pH 8, 600 mM NaCl, 1 mM DTT, 10 mM Glutathione. The elution fractions

were pooled, supplemented with TEV protease and dialysed overnight against the dialysis buffer (20 mM Hepes, 300 mM NaCl, 1 mM DTT). Subsequently, the samples were re-adsorbed onto a Glutathione Sepharose 4 Fast Flow 16/10 column (GE Healthcare) and the flow-through was purified by size exclusion chromatography using a Superdex 200 10/300 GL column (GE Healthcare). Purified samples were stored in gel filtration buffer (20 mM Hepes, 300 mM NaCl, 5 mM DTT). Brat-NHL was expressed and purified as previously described (32).

## Crystallization and structure determination

Mei-P26 NHL crystals were grown at 20°C using the sitting drop vapor diffusion method. The purified protein was concentrated to 11 mg/ml in gel filtration buffer and combined with an equal volume of reservoir solution (0.2 Potassium thiocyanate and 20% PEG3350). After 2 wk of incubation, crystals were fished, cryo-protected with 30% glycerol and subsequently frozen in liquid nitrogen. Datasets were collected at BESSY II Helmholtz-Zentrum Berlin beamlines 14.1 and 14.2. Data processing was performed using XDS (66) and initial phases were obtained by molecular replacement using a search poly-alanine model based of Brat-NHL (PDB code 1Q7F) (28) in Phaser (67). A structure model was built in COOT (68) and subsequently refined in Phenix (69). All dataset and refinement statistics are given in table (Table 1). Model figures were generated using Pymol (70).

## RNAcompete analysis

RNAcompete data for the Brat and Mei-P26 NHL domains were obtained from Prof. Timothy Hughes' laboratory at the University of Toronto (6). The raw data were computationally split into two halves (Set A and B) in the RNAcompete pipeline (35) to facilitate internal data comparisons. Next, the Z-scores for 7-mers for Set A, Set B, and the average of Set A and Set B (Set A + B) were calculated, ranked, and presented on scatter plots. Consensus motifs recognized by Mei-P26 and Brat were calculated from average of two halves of the RNA pool.

## Modeling of Mei-P26-NHL–RNA complexes

The initial structure of the Mei-P26-NHL–RNA complex was generated by superimposing the Mei-P26 protein with the Brat-RNA complex (PDB ID: 5EX7) and copying the coordinates of the Brat RNA ligand. The superposition was performed for the 257 Cα atoms (RMSD = 1.343 Å), which matches between the Mei-26-NHL and Brat proteins. The sequence of that RNA was modified to UUUUUUU, UUUUACA, or UUUGUUGU using UCSF Chimera (71) to prepare three starting structures for Mei-P26-NHL:RNA complexes. Molecular dynamics simulations for Mei-P26-NHL–RNA complexes for the three cases were performed using the Amber18 package (72). Molecular dynamics simulations were run for Brat in complex with UUGUUAA, UUUUACA, and UUGUUGU RNAs as controls. The input structure for the simulation was prepared using tleap in a truncated octahedral box of 10 Å allowance using the TIP3P water model (73). Simulations were performed using the combination of the Amber ff14sb force field for proteins (74) and the χOL3 force field for RNA (75, 76). The structure was energy-minimized for 10,000 cycles with restraints, followed by 10,000 cycles without restraints. The

minimized structures were subjected to heating, density equilibration and short runs of equilibration. The heating was done from 100 K to 300 K for 500 ps with restraints on the entire structure and the density equilibration was performed for 500 ps, also with restraints on the entire structure. The equilibration of the structures was run for four short rounds. The first three rounds of equilibration were run for 200 ps each with the main chain atoms constrained. The final round of equilibration was performed for 2 ns to ensure full convergence and reliability of the models. The production run was run for 1 μs. We have used constant pressure periodic boundary conditions (ntb = 2) with isotropic position scaling (ntp = 1) with a pressure relaxation time taup = 2.0 ps for the production run. The particle-mesh Ewald (PME) procedure (77) was used for computing the electrostatic interactions. The cut-off values used for electrostatics and LJ interactions were set as 12 Å. The equilibration steps were run with the NVT ensemble (ntb = 1), whereas the production run was performed with the NPT ensemble. The minimization was performed using Sander and the subsequent steps were performed using the CUDA version of PMEMD available in the Amber package (78, 79, 80). The simulation trajectories were clustered using the reimplementation of NMRCLUST algorithm (81) available in UCSF Chimera and the representative frames (Figs S4 and S5) are provided as PDB files in the supplementary materials. All simulated models and the used restraints were deposited to a publicly available data repository (Mendeley Data; doi: 10.17632/jvkcfwyz47.1). The Theseus analysis was performed to evaluate the MD simulation by simultaneous superposition of the Brat NHL–UUGUUGU crystal structure (PDB ID: 5EX7) with the Brat NHL–UUGUUGU model and the top five clusters from representative models (82).

## MST

Experiments were conducted using 20 nM of Cy5-labeled oligonucleotides (Sigma-Aldrich) in the buffer containing 20 mM Hepes/KOH, pH 7.5, 100 mM NaCl, 5 mM MgCl$_2$, 5 mM DTT, and 0.0125% Tween 20 for Mei-P26 NHL and 20 mM Tris 8.0, 150 mM NaCl, 1 mM MgCl$_2$, 5 mM DTT, and 0.0125% Tween 20 for Brat NHL. 3:1 serial dilutions of unlabelled protein (starting from the highest concentration of 5 μM) were mixed with labeled oligonucleotide, incubated for 15 min and applied for measurements. Measurements were conducted at 40% MST power and light-emitting diode/excitation power setting 20% in Premium Coated capillaries on the Monolith NT.115 at 25°C (Nanotemper Technologies) (83). Each experiment was performed in at least three replicates. The data were analyzed using the MO.Affinity software (Nanotemper Technologies) at the standard MST on time off 5 s. To calculate dissociation constants ($K_d$), Hill models were fitted to each dataset. The graphs were prepared in GraphPad Prism software. Oligonucleotides used in the experiments are listed in Table S1.

## EMSA

15 pmol of RNA/DNA were radioactively labeled for 1 h at 37°C using 5 U T4 Polynucleotide Kinase (Thermo Fisher Scientific) and 10 μCi γ-$^{32}$P ATP. The reaction was inactivated at 75°C for 10 min and the labeled RNA/DNA was purified by gel filtration (Illustra MicroSpin

G-25 columns; GE Healthcare). The RNAs were diluted to a final concentration of 2 fmol/$\mu$l in a buffer containing 10 mM Tris/Cl pH 7.4, 50 mM KCl, 1 mM EDTA, 1 mM DTT, and 0.4 mg/ml yeast tRNA (Invitrogen). Dilution series of the wild-type Mei-P26 NHL domain and its mutant derivatives were prepared in gel filtration buffer supplemented with 5% glycerol (20 mM Hepes/KOH, pH 7.5, 5 mM DTT, 100 mM NaCl [wt NHL domain] or 300 mM NaCl [mutant derivatives]). 5 $\mu$l of the protein dilution were mixed with an equal volume of the RNA preparation (containing a total of 10 fmol of the labeled RNA) and incubated for 30 min at 4°C before separation by native gel electrophoresis (6% polyacrylamide [37.5:1 acrylamide: bisacrylamide], 5% glycerol, 44.5 mM Tris, and 44.5 mM boric acid) at 230 V for 60 min at 4°C. Gels were dried for 2 h at 80°C and analyzed on a Personal Molecular Imager (Bio-Rad). For RNA stability assays, 10 $\mu$l reactions were set up as described above, containing radioactively labeled RNA and 1,280 nM of the purified NHL domain. Control reactions were supplemented with buffer instead of the protein preparation. After 30 min of incubation at 4°C, RNAs were purified by organic extraction, separated by 15% denaturing PAGE and visualized by autoradiography.

### Thermal shift assays

Recombinant Mei-P26 NHL domain and variants thereof (each at a concentration of 1 g/l) were incubated with SYPRO Orange and 20 mM Hepes, pH 7.5, 300 mM NaCl, 5 mM DTT buffer in 96-well plates followed by centrifugation (5′, 180 rcf). Subsequently, the samples were gradually heated from 4 to 98°C with a rate of 0.2°C/10 s in the CFX96 Real-Time System C1000 Touch Thermal Cycler (Bio-Rad). The fluorescence intensity was measured using an excitation wavelength of 470 nm, whereas monitoring emission at 570 nm.

### Tissue culture

*Drosophila* S2R+ cells were propagated at 25°C at 80% confluency in Express Five SFM supplemented with 10× Glutamax (Thermo Fisher Scientific).

### Western blotting

Cultured cells were harvested and resuspended in lysis buffer (20 mM Tris/Cl, pH 8.0, 150 mM NaCl, 5 mM EDTA, 1% NP-40, and 2% SDS). Protein concentration of cleared lysates was determined using the Bio-Rad protein assay reagent. 25 $\mu$g of total protein were separated by denaturing PAGE and subjected to Western blotting using mouse monoclonal anti-FLAG antibody (M2; Sigma-Aldrich, 1:1,000) followed by probing with an HRP-coupled anti-mouse light chain–specific secondary antibody (1:10,000; Jackson Immuno Research). Detection occurred by using Clarity Western ECL substrate and a ChemiDoc Touch Imaging System (Bio-Rad). After stripping, the membrane was re-probed using mouse anti–$\alpha$-tubulin antibody (DM1A; Sigma-Aldrich, 1:1,000).

### Individual-nucleotide cross-linking and immunoprecipitation (iCLIP)

Cells were transfected for 48 h with 15 $\mu$g DNA per 15-cm dish using Fugene HD (Promega) following the manufacturer's instructions using plasmids encoding full-length FLAG-tagged MeiP26 protein, its NHL domain only, or their respective mutant versions. For identification of the RNA targets, iCLIP2 was used (37). Briefly, a 15-cm dish of cells was washed with PBS and UV-irradiated (120 mJ/cm$^2$ at 254 nm) using a UV Stratalinker 2400 (Stratagene). Next, cell extract was prepared and subjected to RNase treatment using 10 U of RNase I (Ambion). Immunoprecipitation was performed with anti-FLAG antibody (M2; Sigma-Aldrich) or control serum on Dynabeads Protein A (Life Technologies) for 2 h at 4°C. After washing four times with washing buffer (50 mM Tris/Cl, pH 7.4, 1 M NaCl, 0.05% Tween 20), the co-immunoprecipitated RNA was dephosphorylated, ligated to a 3′-RNA linker and 5′-radiolabeled with T4 PNK and [$\gamma$-$^{32}$P]-ATP. Samples were subjected to neutral SDS–PAGE (NuPAGE; Invitrogen) and transferred to a nitrocellulose membrane. Protein/RNA-complexes were visualized by autoradiography. Mei-P26–RNA complexes were cut from the membrane, proteins were digested with Proteinase K and RNA was subjected to iCLIP2 library preparation as previously described (37). Sequencing occurred on a HiSeq 2500 (Illumina). Three independent biological replicates were performed for each protein construct; as a control, non-transfected cells were processed in parallel.

### iCLIP data analysis

The iCLIP data were processed using the iCount software suite and analysis pipeline. The sequencing reads were demultiplexed based on barcodes for individual replicates (allowing one mismatch), PCR duplicates were removed and adapters were trimmed. The reads for each of the replicates were aligned to the *D. melanogaster* genome (ENSEMBL release 98) and processed separately. Cross-linked nucleotides (peaks) were identified and then clustered. Gene loci that produced iCLIP peaks in the experiments conducted with both, the full-length protein and the NHL domain were manually curated. When crosslinking occurred to mitochondrially encoded RNAs (four loci in total) or known contaminants such as snoRNA/scaRNA/snRNA sequences (or similar) present in the host genes (17 loci), the gene loci were excluded from further analyses. Similarly, crosslinking to low complexity regions (mostly A-stretches that resemble polyA-tails) or sequences derived from the transfected plasmids (originating from the Actin 5C promoter or the Mei-P26–coding region) were not considered. In the remaining 214 genes, crosslinking positions were considered equivalent between the individual experiments when the distance between the crosslinking peaks (full-length versus NHL domain) was <50 nt. In these loci, the presence of Mei-P26 RNA target motifs was scored up to 30 nt upstream and 20 nt downstream of the crosslink positions, considering full matches (UUUUACN, UUUUANA, UUUUNCA, or UUUUUUU, 65 loci) or U-rich sequences with four consecutive U residues followed by 3 nt containing at least one additional U residue (63 loci).

### Tethering and reporter assays

For tethering assays, per well of a 96-well plate, 4 × 10$^4$ *Drosophila* SR2+ cells were seeded and transfected using Fugene (Promega) according to the manufacturer's instructions with the following plasmids: (1) 10 ng of a Firefly luciferase reporter with five BoxB

elements in the 3′ UTR (pAC-FL-5boxB, kindly provided by I. Loedige), (2) 30 ng of a control plasmid encoding Renilla luciferase and (3) 60 ng of plasmids encoding either HA- or λNHA-tagged proteins of interest. 48 h after transfection, the cells were lysed with 1× Passive Lysis Buffer (Promega) and luciferase activities were determined using the Dual-Luciferase Reporter Assay System (Promega) and a Centro LB 960 luminometer (Berthold). Relative luciferase units (RLUs) were calculated for each sample by dividing Firefly luciferase activity by Renilla luciferase activity. Activities are expressed as the ratio of RLUs obtained for the tethered, λNHA-tagged proteins relative to the untethered control proteins. Depicted are mean values ± SD of at least three independent biological experiments performed in technical triplicates. 3′ UTR reporter assays were performed analogously. The transfection mixture contained 100 ng of plasmid DNA consisting of 23.3 ng Firefly luciferase reporter plasmid (pHS-FL bearing different 3′ UTR sequences), 1.7 ng Renilla luciferase encoding plasmid (pHS-RL, normalization control) and 75 ng of a pAc5.1 plasmid encoding either 2x FLAG-tagged Mei-P26, or a mutant version thereof. An empty pAc5.1 plasmid served as a control. 48 h after transfection, RLUs were determined as described above and normalized to the empty pAc5.1 vector control. Depicted are mean values ± SD of at least three independent biological experiments performed in technical triplicates.

### Prediction of Mei-P26 structure

To predict spatial structures of Mei-P26 and its parts, corresponding sequences were processed using AlphaFold-multimer (44, 45 Preprint) encased in ColabFold (84 Preprint) package which takes advantage of MMseq2 (85) server for automated MSA generation. In detail, AlphaFold run in the unpaired mode, the number of recycles was set to 3. Side chains were relaxed after prediction using the AMBER force field (86) with default settings.

# Data Availability

The atomic coordinates and respective structure factors for Mei-P26 NHL (PDB ID: 7NYQ) have been validated and deposited at the European Protein Data Bank. Computational models and description of the restraints have been deposited with Mendeley Data (doi: 10.17632/jvkcfwyz47.1). All sequencing data have been deposited at GEO under the following accession number: GSE152013.

# Supplementary Information

# Acknowledgements

We thank the staff at the beamlines 14.1 (BESSY) and P11 (PETRAIII) for their support during data collection and the MCB Structural Biology Core Facility (supported by the TEAM TECH CORE FACILITY/2017-4/6 grant from Foundation for Polish Science) for providing instruments and support. We thank Kate Nie, Debashish Ray, Timothy R Hughes, and Quaid D Morris at the University of Toronto for providing the RNAcompete data and for valuable discussions on the data analysis. We thank Gunter Meister at the University of Regensburg, Inga Loedige at the Max Delbrück Center for Molecular Medicine, Berlin, and Julian König at the Central European Institute of Molecular Biology, Mainz, for sharing reagents. We thank Panagiotis Alexiou at the Central European Institute of Technology, Brno, for discussions on CLIP data analysis. We thank Elizabeth Michalczyk for useful comments on the revised manuscript. This project has received funding from the European Union's Horizon 2020 research and innovation programme under the Marie Skłodowska-Curie grant agreement No. 665778 (M Gaik and A Salerno-Kochan). Computational analyses were performed using the resources of IIMCB, the Poznań Supercomputing and Networking Center at the Institute of Bioorganic Chemistry, Polish Academy of Sciences (grant: 312), the Polish Grid Infrastructure (grants: rnpmd, rnpmc and simcryox), and the Interdisciplinary Centre for Mathematical and Computational Modelling at the University of Warsaw (grants: G73-4 and GB76-30). Funding: National Science Centre, Poland (UMO-2015/19/P/NZ1/02514 to M Gaik, UMO-2019/32/T/NZ1/00420 to A Salerno-Kochan, MAESTRO 2017/26/A/NZ1/01083 to JM Bujnicki). German Research Foundation (SFB960 TP B11, ME4238/1-1 to J Medenbach, RTG2355 325443116 to O Rossbach); German Federal Ministry of Education and Research (01ZX1401D to J Medenbach); Foundation for Polish Science (FNP) (TECH CORE FACILITY/2017-4/6 to S Glatt, TEAM/2016-3/18 to JM Bujnicki); IIMCB statutory funds to JM Bujnicki.

## Author Contributions

A Salerno-Kochan: data curation, formal analysis, funding acquisition, validation, investigation, visualization, methodology, and writing—original draft, review, and editing.
A Horn: data curation, formal analysis, validation, investigation, visualization, and methodology.
P Ghosh: data curation, software, formal analysis, visualization, and methodology.
C Nithin: resources, data curation, software, formal analysis, validation, investigation, visualization, and methodology.
A Kościelniak: formal analysis, investigation, and methodology.
A Meindl: data curation, formal analysis, and investigation.
D Strauss: data curation, formal analysis, investigation, and methodology.
R Krutyhołowa: data curation, software, formal analysis, validation, investigation, and visualization.
O Rossbach: conceptualization, data curation, formal analysis, validation, investigation, and methodology.
JM Bujnicki: data curation, software, formal analysis, supervision, funding acquisition, validation, investigation, and methodology.
M Gaik: conceptualization, data curation, formal analysis, supervision, funding acquisition, validation, investigation, visualization, methodology, and writing—original draft.
J Medenbach: conceptualization, resources, data curation, formal analysis, funding acquisition, validation, investigation, visualization, methodology, project administration, and writing—original draft, review, and editing.
S Glatt: conceptualization, data curation, formal analysis, supervision, funding acquisition, validation, investigation, visualization, methodology, project administration, and writing—original draft, review, and editing.

## Conflict of Interest Statement

The authors declare no competing interests.

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
