## [Reviewer comments · Life Science Alliance]

Life Science Alliance

Molecular insights into RNA recognition and gene regulation by the TRIM-NHL protein Mei-P26

Anna Salerno-Kochan, Andreas Horn, Pritha Ghosh, Chandran Nithin, Anna Kościelniak, Andreas Meindl, Daniela Strauss, Rościsław Krutyhołowa, Oliver Rossbach, Janusz Bujnicki, Monika Gaik, Jan Medenbach, and Sebastian Glatt

DOI: <https://doi.org/10.26508/lsa.202201418>

Corresponding author(s): Sebastian Glatt, Jagiellonian University and Jan Medenbach, University of Regensburg

Review Timeline:

Submission Date:	2022-02-21
Editorial Decision:	2022-02-22
Revision Received:	2022-03-11
Editorial Decision:	2022-04-05
Revision Received:	2022-04-11
Accepted:	2022-04-12

Transaction Report:

February 22, 2022

Re: Life Science Alliance manuscript #LSA-2022-01418-T

Dr. Sebastian Glatt
Jagiellonian University
Malopolska Centre of Biotechnology
Gronostajowa 7a str
Krakow 30-387
Poland

Dear Dr. Glatt,

Thank you for submitting your manuscript entitled "Molecular insights into RNA recognition and gene regulation by the TRIM-NHL protein Mei-P26" to Life Science Alliance. We invite you to re-submit the manuscript, revised according to your Revision Plan.

Thank you for this interesting contribution to Life Science Alliance. We are looking forward to receiving your revised manuscript.

Sincerely,

Eric Sawey, PhD
Executive Editor
Life Science Alliance
<http://www.lsa-journal.org>

B. MANUSCRIPT ORGANIZATION AND FORMATTING:

Point-by-point response

Manuscript number: LSA-2022-01418-T

Corresponding author(s): Monika Gaik, Jan Medenbach and Sebastian Glatt

Reviewer #1 (Evidence, reproducibility and clarity (Required)):

In this paper, Salerno-Kochan et al. determine a high-resolution structure of the NHL domain of Mei-P26, and characterize binding of different linear RNA sequences based on analysis of previously described binding data. The authors find specific RNA sequence recognition by the NHL domain with dissociation constants in the micromolar range, and use MD simulations together with their structure to validate these data and narrow down amino acids important for the association. In addition, they perform iCLIP experiments to determine targets of Mei-P26 in S2 cells, and validate the importance of RNA binding to the regulation of a subset of the targets found. The manuscript is well written and the data was presented carefully. Below I present major and minor points of consideration to address for improving the manuscript.

Major points

- 1) The authors use different NaCl concentrations to perform EMSAs with the WT and mutant Mei-P26 proteins (Methods, 100 mM vs. 300 mM). Seeing as this interaction is expected to be highly dependent on electrostatics, one expects the K_d to vary greatly under different salt conditions. Therefore, the K_d s determined by EMSAs are not comparable.

Response: We have performed additional EMSA experiments with the wt NHL domain at elevated salt concentrations (comparable to the ones used for the mutants). The results show that the binding affinities remain almost unchanged in the presence of different salt concentrations. The additional control assays have been incorporated into Supplementary Figure 3A. We would like to emphasize that the relative difference between the mutants (Fig. 3D) has been observed using identical salt concentrations and all presented MST analyses have been performed using 100 mM NaCl as well.

- 2) The MD experiments focus on systems that bind, but some control MD runs would be helpful in validating the protocol: one with the triple mutant 1150/1172/1175, as well as a run with an RNA sequence not similar to the motifs found. Do the RNA molecules dissociate in both cases?

Response: We prepared an atomic model of Mei-P26 NHL_{R1150A/K1172A/R1175A} and we have performed the extended MD simulations for the mutated variant in combination with UUUUACA and UUUUUU sequences. In the case of Mei-P26 NHL_{R1150A/K1172A/R1175A}, we observed unstable binding of UUUUUUU, where the 5' region binds, unbinds, and rebinds during the course of the simulation. In the case of the UUUUACA sequence the ACA triple nucleotide is not well accommodated by the mutated protein. We added those observations into the main text of the manuscript and provide the respective simulations as supplementary movies.

In addition, MD simulations of wild type Mei-P26 NHL with the original RNA hairpin recognized by Lin41 and a poly-G sequence have been calculated. The poly-G sequence seems to bind to the similar region on the NHL domain as the poly-U sequence, while the Lin41 sequence motif does not fold properly into a hairpin shape prior or during the binding. As observed before for the interaction between Brat NHL and the UUUUACA sequence, the MD simulations might be in some cases biased towards the formation of a complex. In particular, the relative comparison between completely

unrelated sequences might lead to misinterpretations. Foremost, we strongly believe that modelling requires an independent experimental validation. Therefore, we decided to not show the MD simulation for Mei-P26 NHL wt with polyG and Lin41 sequences.

- 3) The method used to assess binding does not seem to allow for saturation of the signal to be reached, however, K_d determination without saturation is extremely unreliable and error-prone, and care should be taken in analysis of the relative K_d s. For example, I would consider K_d s in the range of 1-3 μM to be more or less the same within the experimental error? So the NHL domain of Mei-P26 is not as specific. But the results might be quite different when going to saturation. Then also the weaker binders affinity can be assessed. The authors mention solubility issues. Although understandable, for crystallization the authors could reach 11 mg/ml, which is around 300 μM . This would be more than enough for alternative methods, like ITC to cross-validate RNA binding. Did the authors try?

Response: The MST technique allows the precise quantitative analysis of binding events with a relatively low amount of sample solution. Hence, it was the method of choice to detect the slight differences in dissociation constants between the experimental setups and map the properties of RNA motif that could be bound by Mei-P26 NHL.

We would like to highlight that the MST measurements are performed in thin glass capillaries, which can be coated with additional surface molecules to reduce the unwanted and artificial attachment of proteins to the surface. We tested different types of capillaries, various buffer conditions (e.g. additional of low concentration of detergents) and temperatures to identify optimal experimental settings. After the use of higher concentrations of proteins we experienced a strong decrease in signal at higher concentrations – most likely indicating local precipitation or binding of the protein to the coated glass surface. Therefore, data points at higher concentrations had to be disqualified and excluded from the analyses. It is for these technical reasons that we could not include additional data points at higher concentrations. In our work, we estimated K_d values only for the protein-oligonucleotides pairs for which the binding curves reached saturation. Those experiments were sufficient to validate the computational analysis of RNA-compete data and to define the details of Mei-P26 binding properties, which were then further validated by modelling and iCLIP data. Indeed, we have encountered technical challenges in estimating the K_d for weakly binding sequences, therefore we additionally decided to cross-validate some of the crucial experiments using EMSA approach to be sure that MST measurements are accurate.

- 4) The authors conclude that the RING domain is not important for regulatory activity based on two reporter genes only. Either more genes should be tested, or this conclusion should be more conservative.

Response: We agree that our statement is based on very few observations. Therefore, we have now down-toned the statement on page 28 (line 640) of the revised manuscript – “*The RING domain is not required for the regulation of the analyzed reporter RNAs.*”

- 5) In the introduction the authors write (line 39) that the Brat-NHL domain participates in multiple protein-protein interactions involving factors such as Pum etc., citing a couple of articles, which to my knowledge did not show direct protein-protein interaction between Pum and Brat. It is noteworthy, that a recent article by Macosek et al. (NAR 2021), could demonstrate that no direct interaction between at least the Pumilio homology domain and Brat-NHL domain (nor Nanos-znf domain) exist.

Response: The mentioned sentence has been changed accordingly in the introduction and the latest publications have been added as references.

- 6) The authors expressed the NHL domain in a baculovirus expression system. I wonder if *E. coli* did not work or was that not tried? If it would, then the RNA binding characterization of this domain could have been done in more detail with regards to residue-wise RNA binding propensity, which could validate the MD simulations. A protein-RNA complex structure would of course increase the knowledge and I assume the authors tried to crystallize the complex? Any comments on why this did not work and what has been tried?

Response: We extensively tried to obtain Mei-P26 FL and numerous truncated constructs of the N- and C- terminal domains in different bacterial, yeast and insect expression and co-expression systems. To our regret, only the rather time consuming and expensive insect cell system allowed for obtaining a good quality sample in the amounts suitable to conduct structural, biochemical and biophysical analyses. Therefore, this system had to be used to produce Mei-P26 NHL wild type protein and all mutated variants thereof.

Moreover, we fully agree with the reviewer that information about protein-RNA complex structure would significantly increase the knowledge and we aimed and intensively tried to obtain the co-crystal structure of Mei-P26 NHL bound to RNA. We performed multiple co-crystallization trials using different newly identified Mei-P26 NHL oligonucleotide targets and we tried to identify novel crystallization conditions for the RNA-bound Mei-P26 protein by independent high-throughput screenings. We have collected several complete datasets for crystals that were obtained by co-crystallization with different RNA oligonucleotide sequences. All of the datasets showed almost identical crystal parameters compared to the apo crystals and none of them showed any detectable density for the respective RNA molecules. Therefore, we conclude that all yet identified crystallization conditions counter-selected for conditions that favor protein-RNA complex formation. We shortly mentioned the unsuccessful co-crystallization approach in the discussion section (page 25) "Although none of our numerous attempts yielded co-crystals of the Mei-P26 NHL domain in complex with its RNA substrate, computational modeling allowed us to identify and experimentally validate key amino acid residues involved in RNA recognition." On page 13 of the revised version, we have mentioned that we could express Mei-P26 NHL and its variants only in insect cells and that we did not manage to express the NHL domain or other truncated domain constructs in bacterial expression systems.

Minor points:

- Line 523-525, on what basis were the sequences selected? The authors searched the iCLIP data for RNA sequences obtained from their in vitro analysis. Also, they tested a few further motifs from the data in vitro. Is it possible to in turn extract a consensus motif from the 218 identified mRNA targets computationally and check how it compares with the in vitro data, and if more can be found to go back to in vitro and test another sequence?

Response: The sequences were selected according to the following rationale: to cover a broad selection of features, we selected from the set of genes that exhibited overlapping iCLIP peaks candidates that contained in vicinity to the crosslink position either (a) an identifiable Mei-P26 target motif (eIF4A, Rps20, Rps23, sqd, and Swip-1), (b) a U-rich sequence (bic, chic, and sta), or (c) no such element (Hsp83 and spz). Additionally, all 3' UTR sequences were included for which we had experimentally confirmed interaction with Mei-P26 (Col4A1, Hrb27c, LanA, lost, and Mlc-c). The respective information has been added to the main body of the manuscript.

With the aim to identify the Mei-P26 RNA target motif from the iCLIP data, we performed motif analyses with the individual datasets obtained from either the full-length protein or the NHL domain. It is important to stress that the cross-link position in the RNA does not necessarily lie within the respective binding motif, which complicates and impedes motif identification. Our analyses therefore encompassed short RNA regions flanking the respective crosslink positions (see Methods section). With this approach, we could identify an enrichment of U-rich RNA sequences in proximity of the crosslink position.

However, this was only one of several motifs that were statistically enriched. iCLIP experimentation is known to be prone for false positive hits, particularly when employing over-expressed proteins. We are therefore reluctant to conclude particular sequence motifs bound by the protein from the data. For the same reasoning, we refrain from claiming that the identified, U-rich sequence motif is 'the correct one' - this would clearly be a conclusion that is biased by the results of the *in vitro* binding studies. Also, this would partially ignore the experimental diversity that is typical for this kind of methodological approaches.

- Line 586-588. Actually, the affinity is very similar to Brat-NHL for 6-mer RNAs (around 1 μ M), see also Macosek et al., 2021, NAR.

Response: The information and reference have been added to the respective sections of the main text.

- Figure 2D presents a K_d for BRAT1 with UUUUUAA with an error of 0, and in general the errors seem underestimated. Please check error analysis method.

Response: Following the comment, we again carefully checked the error analyses. All K_d s were calculated as an averaged value from at least 3 independent experiments. The employed MST approach gives highly reproducible results for the experiments in which oligonucleotide fragments are bound to protein. The indicated error of UUUUUAA has been double checked and the precise near-zero error has been indicated in the updated figure. In general, the error increases with the lower binding affinities, but it is indeed relatively small.

- The final model of the paper could be shown with an AlphaFold2-predicted structural model to better give the reader a sense of the relative domain sizes of full-length Mei-P26. On that front, out of interest, how does their crystal structure compares to the AF2 structure?

Response: We complemented our study with modelling of full-length Mei-P26 protein and its NHL domain by AlphaFold 2, which is already included in the tentatively revised version of the manuscript (**Supplementary Fig. 12**). To directly answer the question of the reviewer, we compared the experimentally determined Mei-P26 NHL crystal structure and the AF2 model of NHL - they are almost identical with an RMSD of 0.6 \AA^2 for 273 aligned amino acids. The model based on the full-length sequence of the Mei-P26 protein shows a largely unstructured N-terminus (S-rich region) and central part of the protein (Q-rich region), but a compact and folded BB-RING-CC motif and the NHL domain. As certain TRIM-NHL proteins have been suspected to dimerize, we also for the first time modelled a potential homodimer of Mei-P26 using a customized AF2-multimer approach. We separately modelled the N- and C-terminal regions of Mei-P26, in both cases including the suspected dimerization module in the previously predicted coiled-coil (CC) domain. Strikingly, AF2 predicts the dimerization along an unprecedented helix with high confidence. Though, the relatively long helix is located between the N-terminal BB-RING-CC motif and the CC domains. We added paragraphs about Mei-P26 dimerization prediction in result and discussion sections and we incorporated an additional supplementary figure to the revised version of the manuscript. Furthermore, we detected the presence of similar helices

and their dimerization potential in Brat, Lin41/Trim71 and Abba/Thin and added a comment about the potential presence of this novel feature to the new paragraph. In addition, we present the analyses of the other members of the TRIM-NHL protein family in the newly added panel D in Supplementary Fig. 12.

- Avidity is not well-defined

Response: We aimed to provide a concise description of the avidity effect as we present it as a possible explanation for our observations, but we cannot ultimately measure its absolute contribution to the effect. We explain it as “This effect describes a scenario where an individual binding event increases the probability of additional interactions occurring in its close proximity.” In the revised manuscript we have compared it to the concept of affinity to explain and highlight the conceptual difference.

- Could the authors give some possible explanations for the discrepancy between the sequence analysis to find the best RNA binders, and the MD? In other words, why is SEQ3 the best binder according to MD?

Response: We describe these differences already in the original text (page 16,17 lines 356-370) “According to our analysis, SEQ3 binds most stably to the Mei-P26 NHL domain and converges into a well-defined conformation (Fig. 2B, Supplementary Fig. 4). Simulations with the top-scored polyU oligonucleotide (SEQ1) and the inverted SEQ3 sequence (ACAUUUU) resulted in much higher RMSD values, indicating a weaker fit in comparison to the SEQ3 RNA (Supplementary Fig. 4A). The top five clusters obtained by the MD simulations with SEQ3 are much more similar to each other than any of them compared to the clusters obtained with SEQ1. Moreover, we observed lower variance in the 3’ region and more structural heterogeneity of the 5’-RNA docking site between the individual clusters of the Mei-P26:UUUUACA models (Supplementary Fig. 4B). Together, our simulations emphasize the importance of the ACA trinucleotide anchor adjacent to poly-uridine stretches for Mei-P26 RNA recognition (Fig. 2B, Supplementary Fig. 4). Although similar affinities were determined for both oligonucleotides (Fig. 2C, Supplementary Fig. 6), an elevated value for the polyU could have been indirectly caused by the so-called avidity effect (57) (Fig. 2D). This effect describes a scenario where an individual binding event increases the probability of additional interactions occurring in its close proximity.” It is hard to speculate about the specific reasons for the differences between experimental binding studies and simulations. Therefore, we hope the provided section is sufficient for the reviewer.

- The discussion section would benefit from more in-depth analysis of the data presented; at the moment it reads mostly as a summary of the paper, and some of the points are not very clearly made. Below are some suggestions:
 - Is there a functional link between the identified regulated mRNAs, besides the very general functions listed? How does that relate to the known roles of Mei-P26?

Response: We previously touched on these rather speculative issues in the discussion section - “Despite these limitations, our experimentation allowed us to identify numerous transcripts that encode translation factors and ribosomal proteins among the Mei-P26 target mRNAs. Previously, Mei-P26, like Brat, has been broadly implicated in the regulation of ribosome biogenesis by controlling the expression of Myc (21, 79, 80), which stimulates the expression of the Pol I transcriptional machinery thus promoting ribosome biogenesis (21, 81, 82). The coordinate regulation of ribosomal proteins and ribosomal RNA (rRNA) synthesis by Mei-P26 might allow it to efficiently tune ribosome biosynthesis which is linked to cell growth and the switch between proliferation and differentiation.” We have now followed the reviewer’s suggestion to slightly expand this section in the discussion.

- Could the authors expand on the observation that Mei-P26 can activate some genes in an RNA-binding dependent manner?

Response: At this point the mechanisms remain elusive, but we previously speculated about the Mei-P26 dependent recruitment of activators to the respective RNA. This is also highlighted in Figure 5E – “How can these two seemingly different activities be explained? In tethering experiments, where the closely related protein Brat acts as a strong silencer of gene expression, Mei-P26 exhibits only weak activity. This suggests that either tethering disturbs its function or Mei-P26 itself is not a strong regulator of gene expression. In the latter scenario, Mei-P26 might function in promoting complex assembly, recruiting other factors that act in gene regulation. Previously, Ago1, Sxl, Bam, Bgcn, and Wuho have been identified as co-repressors that act in concert with Mei-P26 in post-transcriptional regulation of gene expression (Fig. 5E) (18, 19, 21, 30, 74). Potential activators that can be recruited by Mei-P26 for gene regulation remain to be identified.”

- References are missing throughout the paper, in particular those referring to previously published methods. The companies providing certain reagents are also missing.

Response: The references and reagent providers have been added to the method section in the revised version of the manuscript.

- Thermal shift assays: specify g force used during centrifugation and not r.p.m.

Response: g force has been specified in the method section of the revised version of the manuscript.

- Provide an unambiguous gene code for the full-length protein used, as well as what boundaries were used to remove the N-terminal RING domain. In Supplemental Figure 11, there appears to be a construct named ΔN , but this is not defined.

Response: We unified the naming of the full-length protein and adjusted the labels in Suppl. Figure 11 in the revised version of the manuscript.

- MST assays: no pH specified for Mei-P26

Response: MST measurements have been performed using a HEPES/KOH buffer at pH 7.5. We have now specified the pH for the MST measurements in the respective Materials and Methods section.

- With the data already obtained, is it possible to determine whether the FL and NHL-only versions of Mei-P26 have distinct RNA-recognition motifs? If so, what could contribute to additional sequence requirements?

Response: As shown in Supplementary Fig. 9B and explained on pages 25/26 - “We provide experimental evidence (Supplementary Fig. 9B) that in Mei-P26 protein regions outside the NHL domain also contribute to RNA target recognition/binding. *In vivo*, it is most likely that additional protein partners contribute to stable complex formation between Mei-P26 and its RNA targets.” As shown, the RKR mutation leads to a much less dramatic reduction of RNA binding in the full-length protein than in the NHL domain alone. We also show that the respective mutation has a strong effect on the recognition of a specific RNA motif. Hence, we present some direct evidence that the regions outside of the NHL contribute to the RNA binding, but we cannot exclude that these regions recognize the same motifs – even if we believe that these regions bind RNA motifs close to the NHL domain.

- line 285, proftein should be protein

Response: The typo has been corrected.

- line 292, the authors call it a solvent channel, which sounds like an annotated function to the domain. Has it anywhere shown that this hole serves as a solvent channel? If so, please cite this work.

Response: To our knowledge, there is no functional data showing that the NHL domain acts as a channel. The description and term have been changed and rephrased in the revised version.

Other suggestions for improving the manuscript:

- The biological relevance of the findings of the paper are not clearly shown, given that the experiments performed do not take place in a context where the relevance of RNA binding to Mei-P26 function in *Drosophila* can be tested. Such experiments would greatly strengthen the study.

Response: We fully agree with the reviewer, but given the typical time frame required for careful genetic analyses in *Drosophila* (including all relevant controls such as e.g. rescue experiments of loss-of-function mutants), the suggested work is well beyond the scope of a typical revision.

Reviewer #1 (Significance (Required)):

This manuscript provides interesting insights into the RNA recognition of Mei-P26 via its NHL domain. Although altogether technically sound, the insights would be more reliable with an atomic resolution structure and of course in context of the mRNP environment, which is difficult to obtain. This study adds information about the NHL domain in general and how these recognize RNA. Thus, this study is of interest mostly to researchers working with a similar domain in relation to RNA binding, or which work with Mei-P26 and its biological implications.

Reviewer #2 (Evidence, reproducibility and clarity (Required)):

Mei-p26, a TRIM-NHL protein, controls stem cell maintenance and differentiation. In this manuscript, the authors (1) examined the crystal structure of Mei-p26 NHL domain, (2) characterized (2) the critical amino acids that control the specificity of binding with single strain RNA, and potential sequences of the single strain RNA that can bind with TRIM-NHL, (3) showed different binding preference between Mei-p26 NHL domain and its closely related TRIM-NHL proteins, Brat. Lastly, they identified the potential targets of Mei-p26 in the cultured S2 cells, and show those RNA expressions are either upregulation or downregulation when NHL domain of Mei-p26 is disrupted.

Major comments

1. The in vitro data are clear and convincing. However, several important results are left in the supplementary information, making it difficult in reading through the manuscript. The author might consider moving some critical supplementary information back to the main figures.

Response: We believe that the main figures nicely show the main findings and that the majority of supplementary information is rather supportive. We would like to keep the current number of main figures, which are information-dense and data-rich. Therefore, we would rather avoid major rearrangements for the figures/figure panels and would not like to introduce an additional main figure.

2. The S2 cell has a relatively lower expression of Mei-p26 as stated by the author. In addition, roles of Mei-p26 is critical in stem cell rather than in differentiated somatic cells. Therefore, expressions of potential targets of Mei-p26 should be tested in the fly tissue/organ (eg.

ovaries) carrying mei-p26 NHL mutation. If the line is not able to obtain, then test in mie-p26 mutant flies with or without Mei-p26 NHL domain expression.

Response: We agree with the reviewer that analyses in tissues/organs of animals would be very revealing. However, the time frame for genetic experimentation aimed at addressing these questions is well beyond the scope of a typical revision (see answer to reviewer 1 issue 3).

Minor points

- The manuscript is somehow difficult for me to flow through.

Response: The manuscript was revised by a native speaker before submission – the revised version has been checked again before resubmission.

Reviewer #2 (Significance (Required)):

Mei-p26 is a key posttranscriptional regulator, while how its NHL recognizes and selects RNA targets for regulation remains unclear. The author's works greatly add knowledge to the molecular feature of Mei-p26 NHL domain.

Reviewer #3 (Evidence, reproducibility and clarity (Required)):

Summary

The authors here present a comprehensive study in RNA recognition by Mei-P26, a TRIM-NHL protein. TRIM-NHL family of RNA binding proteins pose quite a challenge in studying them as changes in few amino acids on the top surface of NHL domain leads to a change in the consensus RNA sequence/structure that they recognize. In addition they are pleiotropic in functionality and hence a detailed study like here is required to understand how these proteins recognize and regulate RNA. The authors solve a high resolution crystal structure of the NHL domain of Mei-P26 and perform a series of modelling experiments to understand its specificity in RNA binding. They support the modelling data with in vitro binding assays. At the end they perform iCLIP to identify RNA targets of Mei-P26 using *Drosophila* S2 cells and use reporter assays to analyze RNA regulation via the NHL domain of Mei-P26. While the binding specificity of Mei-P26 has been reported using in-vitro assays like RNA compete the authors here identify the specific amino acids that are required to recognize different RNA sequences. The functional experiments are performed in S2 cells and not in the *Drosophila* germline, therefore limiting the relevance of the RNA targets identified. Overall, this a detailed study of RNA binding specificity of Mei-P26 with some data to understand the downstream RNA regulation. This reviewer recommends the publication of this manuscript once the following comments are addressed.

Major comments

1. The binding affinity of Mei-P26 towards cognate RNA motifs in in-vitro binding assay is in low micromolar range. Do the RNA targets have more than one cognate binding sequence in their 3'UTR. Does the number of binding motifs increase the binding affinity? Disulphide bond between Cys1030 leads to dimer formation in vitro. The authors should comment if the protein dimerizes in vivo and if that has an effect on its RNA binding. This should also be explained in context of different sets of RNAs pulled-down by full-length Mei-P26 and just the NHL domain. A conclusive experiment could be to check binding with a mutation in Cy1030 in the NHL domain of the protein is stable and can be purified.

Response: The low micromolar range of Kd is the typical value for RNA binding domains (RBDs) alone and also for known NHL domains that interact with RNA (eg. Lin41 NHL - Kumari et. al 2018, Brat NHL - Macosek et al., 2021). To reach higher specificity proteins use multiple RBDs or form higher

macromolecular complexes. In the case of Mei-P26, both scenarios are equally possible. The newly performed Alpha Fold 2 predictions of Mei-P26 full-length protein showed that Mei-P26 could utilize its N-terminal domain to dimerize and several molecular partners were already characterized for this protein. Cys1030 in the NHL domain most likely is an artifact of the crystallization approach. We have analyzed the interaction surface between the two monomers around Cys1030 using the *Consurf* server and we spotted very low conservation of the amino acids. We also used AF2-multimer to predict homodimers of Mei-P26, but did not observe any indications that the two NHL come close to each other or interact with each other via the region around Cys1030. These observations indicate that there is a small chance that dimerization observed *in vitro* has any biological meaning.

2. Is the sequence of NHL domain used in crystallization/*in vitro* binding experiments is exactly the same as that in the iCLIP experiment? Please indicate this in both methods and results section. In case a different sequence is used, please clarify.

Response: The same sequence with the same domain borders (908-1206, protein isoform C) was used for the NHL in the *in vitro* binding and iCLIP experiments. In contrast, in the previous studies (Loedige et al. 2015), which were the basis for the reanalyzed RNAcompete data, a slightly longer NHL from Mei-P26 isoform A (NHL₈₉₉₋₁₁₈₉) construct that was still fused to a GST-tag was used. We have added information on the consistency between the used NHL domain constructs in the revised manuscript and add the description in the methods section.

3. Figure 4D, Page 23 - Are the 502 mRNAs bound by full length Mei-P26 enriched with bonafide target sequences in their 3'UTRs? How does the frequency of bonafide target sequences differ (if at all) between the 3 classes - bound by Full length protein only, NHL domain only and by both. Going back to comment 1, how many target sequences are their on the UTRs' of these targets?

Response: U-rich motifs of 7 nts in length are widely distributed throughout the transcriptome and encountered in approx. 95% of the *Drosophila* 3' UTRs. The RNAs crosslinked to the proteins do not show an overall enrichment of the respective motifs. However, we detect a local enrichment of U-rich motifs in vicinity of the crosslink site in the iCLIP experiments (20 nt upstream and 10 nt downstream of the X-link position) where the respective motifs occur with a 9-fold increased frequency. It is striking to us that Mei-P26 does not show a detectable association with most of its bona fide target sequence motifs present in the transcriptome. At the current stage, we can only speculate that additional factors (such as dimerization or interaction partners) contribute to its stable RNA association.

4. As Mei-P26 binds to mRNA directly, the tethering assay can be misleading as the binding affinity in tethering assay can be very different from the direct binding.

Response: Indeed, the affinity between the RNA-binding domain and its RNA target sequence that we have used in the tethering assay is much higher than the affinity of Mei-P26 itself to RNA. In the tethering assay, we aim to robustly recruit Mei-P26 to the reporter RNA to determine its regulatory effect, bypassing the requirement for RNA binding via its NHL domain. Such experimentation can be very revealing and e.g. allow the dissection of domain requirements for regulation. Unfortunately, and in contrast to another closely related TRIM-NHL protein, Brat, Mei-P26 exhibits no regulatory activity upon tethering (see Fig. 5A). Various scenarios can account for this, e.g. that interactions that are important for regulation are disturbed or cannot occur upon tethering, which has previously been reported for e.g. Sex-lethal (Grskovic et al., 2003). In addition to tethering-based experimentation, we present additional functional analyses based on binding of Mei-P26 to reporter mRNAs via the identified U-rich sequence elements (Figure 5B-D).

Minor comments

1. The different protein structures in the Sup Fig 1D should be labelled in the figure.

Response: The labels have been added to the Suppl Fig. 1D.

2. Lin41 binds to the loop region of a specific stem-loop motif. Calling it just dsRNA is misleading in Figure 1B-C, page 15.

Response: The label has been changed and the sequence is called "Lin41 SEQ".

3. Sup figure 3C is discussed before Sup figure 3A-B. This figure should be reorganized.

Response: The order of the figure has been reorganized in the revised version of the manuscript.

4. Page 16, line 347 - using the word "flanked" is ambiguous as it could be either/both 5' and 3' of the said U stretch. The authors should clarify that the additional A/C residues are 3' to the uridines.

Response: The sentence has been rephrased according to the Reviewer's suggestion.

5. Page 17, line 371 - use of the phrase "RNA recognition modes" is ambiguous. The authors should explain this.

Response: The sentence has been rephrased in the revised version of the manuscript.

6. Page 26, line 585 - "Our measured affinities...." - of what? This sentence is ambiguous.

Response: The sentence has been rephrased.

Reviewer #3 (Significance (Required)):

TRIM-NHL family of RNA binding proteins pose quite a challenge in studying them as changes in few amino acids on the top surface of NHL domain leads to a change in the consensus RNA sequence/structure that they recognize. In addition they are pleiotropic in functionality and hence a detailed study like here is required to understand how these proteins recognize and regulate RNA. It is an important family of proteins having roles in proliferation and differentiation. Such studies are critical for dissecting their function during development and in diseases.

Having worked with RNA binding proteins and studying their roles in development and diseases, I can say that the researchers working with TRIM-NHL proteins and other families of RNA binding proteins with common challenges will appreciate this work.

April 5, 2022

RE: Life Science Alliance Manuscript #LSA-2022-01418-TR

Dr. Sebastian Glatt
Jagiellonian University
Malopolska Centre of Biotechnology
Gronostajowa 7a str
Krakow 30-387
Poland

Dear Dr. Glatt,

Thank you for submitting your revised manuscript entitled "Molecular insights into RNA recognition and gene regulation by the TRIM-NHL protein Mei-P26". We would be happy to publish your paper in Life Science Alliance pending final revisions necessary to meet our formatting guidelines.

- There is only one panel in Figure S2, so please remove the panel label in the figure
- Author Contributions: there is no specific mention of authors AM or RK

A. FINAL FILES:

B. MANUSCRIPT ORGANIZATION AND FORMATTING:

****It is Life Science Alliance policy that if requested, original data images must be made available to the editors. Failure to provide**

original images upon request will result in unavoidable delays in publication. Please ensure that you have access to all original data images prior to final submission.**

The license to publish form must be signed before your manuscript can be sent to production. A link to the electronic license to publish form will be sent to the corresponding author only. Please take a moment to check your funder requirements.

Sincerely,

Reviewer #1 (Comments to the Authors (Required)):

In this paper, Salerno-Kochan et al. determine a high-resolution structure of the NHL domain of Mei-P26, and characterize binding of different linear RNA sequences based on analysis of previously described binding data. The authors find specific RNA sequence recognition by the NHL domain with dissociation constants in the micromolar range, and use MD simulations together with their structure to validate these data and narrow down amino acids important for the association. In addition, they perform iCLIP experiments to determine targets of Mei-P26 in S2 cells, and validate the importance of RNA binding to the regulation of a subset of the targets found. The manuscript is well written and the data was presented carefully. The authors also carefully addressed the concerns I had with the original version to my satisfaction almost completely. I only want to raise the following:

My original point:

The authors use different NaCl concentrations to perform EMSAs with the WT and mutant Mei-P26 proteins (Methods, 100 mM vs. 300 mM). Seeing as this interaction is expected to be highly dependent on electrostatics, one expects the Kd to vary greatly under different salt conditions. Therefore, the Kds determined by EMSAs are not comparable.

Author's response:

We have performed additional EMSA experiments with the wt NHL domain at elevated salt concentrations (comparable to the ones used for the mutants). The results show that the binding affinities remain almost unchanged in the presence of different salt concentrations. The additional control assays have been incorporated into Supplementary Figure 3A. We would like to emphasize that the relative difference between the mutants (Fig. 3D) has been observed using identical salt concentrations and all presented MST analyses have been performed using 100 mM NaCl as well.

My new response:

Response to authors: In supplemental Figure 3A, RNA binding by the WT NHL domain was re-assessed with 150 mM NaCl (higher than the original 100 mM). The figure shows that there is still binding of RNA, although it is difficult to directly compare with the original data in Figure 3D. It's also unclear whether at the 300 mM NaCl concentration used in the case of the NHL domain mutants, the WT version would still bind. Nevertheless, the authors point out that the series of mutant NHL domains were assessed under identical conditions, which shows that increasing the number of mutations disrupts binding further, and that the MST data was carried out under identical buffer conditions for all protein versions.

Reviewer #3 (Comments to the Authors (Required)):

The authors have addressed the comments satisfactorily and I support the publication of this manuscript.

April 12, 2022

RE: Life Science Alliance Manuscript #LSA-2022-01418-TRR

Dr. Sebastian Glatt
Jagiellonian University
Malopolska Centre of Biotechnology
Gronostajowa 7a str
Krakow 30-387
Poland

Dear Dr. Glatt,

Thank you for submitting your Research Article entitled "Molecular insights into RNA recognition and gene regulation by the TRIM-NHL protein Mei-P26". It is a pleasure to let you know that your manuscript is now accepted for publication in Life Science Alliance. Congratulations on this interesting work.

DISTRIBUTION OF MATERIALS:

Again, congratulations on a very nice paper. I hope you found the review process to be constructive and are pleased with how the manuscript was handled editorially. We look forward to future exciting submissions from your lab.

Sincerely,
